# A gas-to-particle conversion mechanism helps to explain atmospheric particle formation through clustering of iodine oxides

Juan Carlos Gómez Martín [1]✉, Thomas R. Lewis[2,3], Mark A. Blitz[3], John M. C. Plane[3], Manoj Kumar[4], Joseph S. Francisco[4] & Alfonso Saiz-Lopez [2]✉

Emitted from the oceans, iodine-bearing molecules are ubiquitous in the atmosphere and a source of new atmospheric aerosol particles of potentially global significance. However, its inclusion in atmospheric models is hindered by a lack of understanding of the first steps of the photochemical gas-to-particle conversion mechanism. Our laboratory results show that under a high humidity and low $HO_x$ regime, the recently proposed nucleating molecule (iodic acid, $HOIO_2$) does not form rapidly enough, and gas-to-particle conversion proceeds by clustering of iodine oxides ($I_xO_y$), albeit at slower rates than under dryer conditions. Moreover, we show experimentally that gas-phase $HOIO_2$ is not necessary for the formation of $HOIO_2$-containing particles. These insights help to explain new particle formation in the relatively dry polar regions and, more generally, provide for the first time a thermochemically feasible molecular mechanism from ocean iodine emissions to atmospheric particles that is currently missing in model calculations of aerosol radiative forcing.

[1] Instituto de Astrofísica de Andalucía, CSIC, 18008 Granada, Spain. [2] Department of Atmospheric Chemistry and Climate, Institute of Physical Chemistry Rocasolano, CSIC, 28006 Madrid, Spain. [3] School of Chemistry, University of Leeds, LS2 9JT Leeds, UK. [4] Department of Earth and Environmental Science and Department of Chemistry, University of Pennsylvania, Philadelphia, PA 19104-6323, USA. ✉email: jcgomez@iaa.es; a.saiz@csic.es

The impacts of iodine on tropospheric and stratospheric chemistry were anticipated by seminal work published in the 80s and 90s[1,2]. The characterization of global iodine sources[3,4] and the demonstration of its ubiquitous presence in the marine boundary layer (MBL)[5,6], the free troposphere[7,8], and the stratosphere[9] have enabled an assessment of the important role of iodine in $O_3$ depletion in past, present, and future climate scenarios[5,9–15]. Iodine oxide-driven new particle formation was first observed in mid-latitude coastal locations[16–18]. Its high nucleation and growth rates may enable new iodine particles to survive fast scavenging by background aerosol[19] and to develop into sea mist and fog[20]. Iodine oxide particles (IOPs) have also been observed in the polar MBL[21–23]. Field observations in diverse locations with completely different meteorological and chemical conditions suggest a wider relevance of this phenomenon in the global atmosphere.

Laboratory studies have investigated the effect of different environmental variables on nm-sized IOPs[17,24–26]. For example, both high temperature and humidity slow down IOP growth and reduce IOP number density relative to colder/drier conditions[24,26]. The $I_2O_3$ and $I_2O_4$ molecules have been observed in the laboratory and linked to IOP formation in dry conditions[27]. However, the O/I ratio measured in the particles collected has been shown to be closer to $I_2O_5$[26]. $I_2O_5$ is the anhydride form of iodic acid ($HOIO_2$), which is the proposed composition of liquid IOPs in the humid MBL[28]. Recently, field observations of $IO_3^-$ and $HIO_3^-$ containing cluster anions by chemical ionization–atmospheric pressure interface–time-of-flight mass spectrometry (CI-API-ToF-MS) have been reported[23]. These observations, which employ nitrate ($NO_3^-$) and acetate ($CH_3COO^-$) reagent ions for reaction with the analytes, have been interpreted as resulting from atmospheric gas-phase $HOIO_2$ and molecular cluster formation via $HOIO_2$ addition steps. Well-established gas-phase kinetics and thermochemistry do not indicate any straightforward route to $HOIO_2$. The $OIO + OH$ reaction is predicted to yield $HOIO_2$[29], but the concentration of OH in the MBL is far too low to explain IOP formation. It has been proposed that water plays an important role in generating $HOIO_2$ and IOPs[23,30], although a strong anticorrelation between particle production probability and water vapor has been observed in the field[31]. In order to form $HOIO_2$, $H_2O$ must be able to intercept some of the initial products of the reaction between atomic iodine and $O_3$, i.e., $IO_x$ (I, IO, and OIO) or $I_2O_y$. The bimolecular reactions of $IO_x$ with $H_2O$ are precluded by their large endothermicities, and termolecular reactions form only weakly bound clusters[32]. A reaction of any of these $H_2O…IO_x$ adducts with $O_3$ to form oxyacids requires substantial bond rearrangement and barriers are expected. Recent ab initio studies have found a substantial barrier in the potential energy surface (PES) of the $I_2O_5 + H_2O \leftrightarrow 2HOIO_2$ reaction[30,33], precluding not only formation of $HOIO_2$ but also a nucleation mechanism depending on the formation of $I_2O_5$ from the $HOIO_2$ self-reaction[23,30].

Here, we present results from flow tube kinetic experiments in order to fill the knowledge gap between gas-phase atmospheric iodine molecules and IOPs. Our results show that water reacts very slowly with atomic iodine and iodine oxides, and that forming particulate $HOIO_2$ does not require gas-phase $HOIO_2$, implying a limited role of oxyacids in IOP formation, which is instead initiated by clustering of $I_xO_y$. Based on our experiments and calculations, an iodine gas-to-particle conversion mechanism is proposed.

## Results

**Pulsed laser photolysis experiments**. We have carried out laboratory flow tube kinetic experiments to look at IOP formation and evaporation (see "Methods" and Supplementary

Figs. 1 and 2) starting from standard chemistry (Supplementary Table 1). Figure 1a shows an average of mass spectra recorded for time delays of up to 20 ms between 248 nm pulsed laser photolysis (PLP) of a mixture of $O_3$ and $I_2$ at 10 Torr of $N_2$, and 10.5 eV photoionization (PI). This "dry" spectrum shows positive ion peaks generated by near-threshold PI (Supplementary Table 2) of known neutral iodine species: $I^+$ ($m/z = 127$), $IO^+$ (143), $OIO^+$ (159), $I_2^+$ (254), $I_2O^+$ (270), $I_2O_2^+$ (286), $I_2O_3^+$ (302), and $I_2O_4^+$ (318). A peak corresponding to $I_2O_5^+$ ($m/z = 334$) is absent. The predicted vertical ionization potential of $I_2O_5$ is 11.4 eV, but this molecule is not observed with the 11.6 eV PI beam either, as shown in Fig. 2. The major peaks of the higher m/z progressions observed in Fig. 1a are $I_3O_7^+$ (493), $I_5O_{12}^+$ (827), $I_7O_{17}^+$ (1161), $I_9O_{22}^+$ (1495), $I_{11}O_{27}^+$ (1829), and $I_{13}O_{32}^+$ (2163). Minor peaks which indicate up to four missing oxygen atoms are also observed. Starting from $I_3O_7^+$ and moving to higher m/z, the peaks are separated by $I_2O_5$ units (Fig. 1b), even though gas-phase $I_2O_5$ does not appear to form. The chemical composition of the main and last peak of each progression can be described as $(I_2O_5)_nOIO^+$. Molecular clusters with even numbers of iodine atoms ($I_4O_7^+$, $I_4O_9^+$) are also detected but are very minor.

A succession of peaks in a mass spectrum does not necessarily reveal how a nucleation mechanism works. Laboratory time-resolved multiplexed experiments (Fig. 3) provide the sequence of reactions and are therefore essential to interpret mass spectra obtained in the field or in the chamber or flow tube steady-state experiments. The initial growth of the concentration versus time curves is generally faster for smaller iodine-bearing molecules, and the peak concentrations are reached in the following order: $IO \rightarrow OIO \rightarrow I_2O_3 \rightarrow I_2O_4 \rightarrow I_3O_7 \rightarrow I_5O_{12} \rightarrow I_7O_{17} \rightarrow I_9O_{22}$, etc. (see kinetic analysis in Fig. 4). The kinetic traces indicate that $I_2O_y$ form $I_3O_y$ species preferentially over $I_4O_y$ adducts ($I_4O_y$ photofragmentation to form $I_3O_y^+ + IO_x$ would occur well above 10.5 eV, Supplementary Table 2). At longer time delays between PLP and PI, all peaks with the number of iodine atoms $x \leq 2$ show kinetic growth at a similar rate compared with the larger clusters, indicating that the latter fragment into smaller constituent subunits following PI. This growth occurs at earlier delay times when the initial concentration of IO is higher (higher excimer laser energy or higher $O_3$ concentration, Supplementary Fig. 3) and/or the PI energy is higher. Because of the second-order reactions involved, starting with the IO self-reaction[34], the formation and removal time scales of $I_xO_y$ depend on the initial concentration of IO, which itself depends on the fraction of $O_3$ photolyzed. Thus, for high IO, $I_xO_y$ clustering is fast and a dominant sink compared to other slow reactions.

Addition of water (mixing ratio $x(H_2O) = 2\%$) at 10 Torr results in a limited reduction of the $OIO^+$ and $I_xO_y^+$ signals (Fig. 1c). No reaction products are detected, except for the peak at $m/z = 144$, i.e., $HOI^+$; this results from OH scavenging by $I_2$, which also prevents any further $HO_x$ chemistry. The mass peak progressions in the presence of water remain the same: neither $HOIO_2$ nor $HOIO_2$-containing peaks (e.g., $(I_2O_5)_n(HIO_3)_{0-2}$[23]) are observed, and no clusters with an even number of iodine atoms emerge. The calculated ionization potential of $HOIO_2$ is 11.3 eV, yet again no signal is observed on a time scale of 4 ms when the PI beam is tuned to 11.6 eV (Fig. 2). Regarding larger oxyacid-containing clusters, the PI threshold is expected to decrease as the size of a cluster increases, and therefore they should photoionize or even photofragment at 10.5 eV. Thus, any hypothetical $I_xO_y + H_2O$ reaction forming $HOIO_2$ occurs at a slower rate than the $I_xO_y$ clustering reactions under these conditions. Given the lack of reaction products upon addition of water, the limited decrease of the $I_xO_y$ signals indicates that

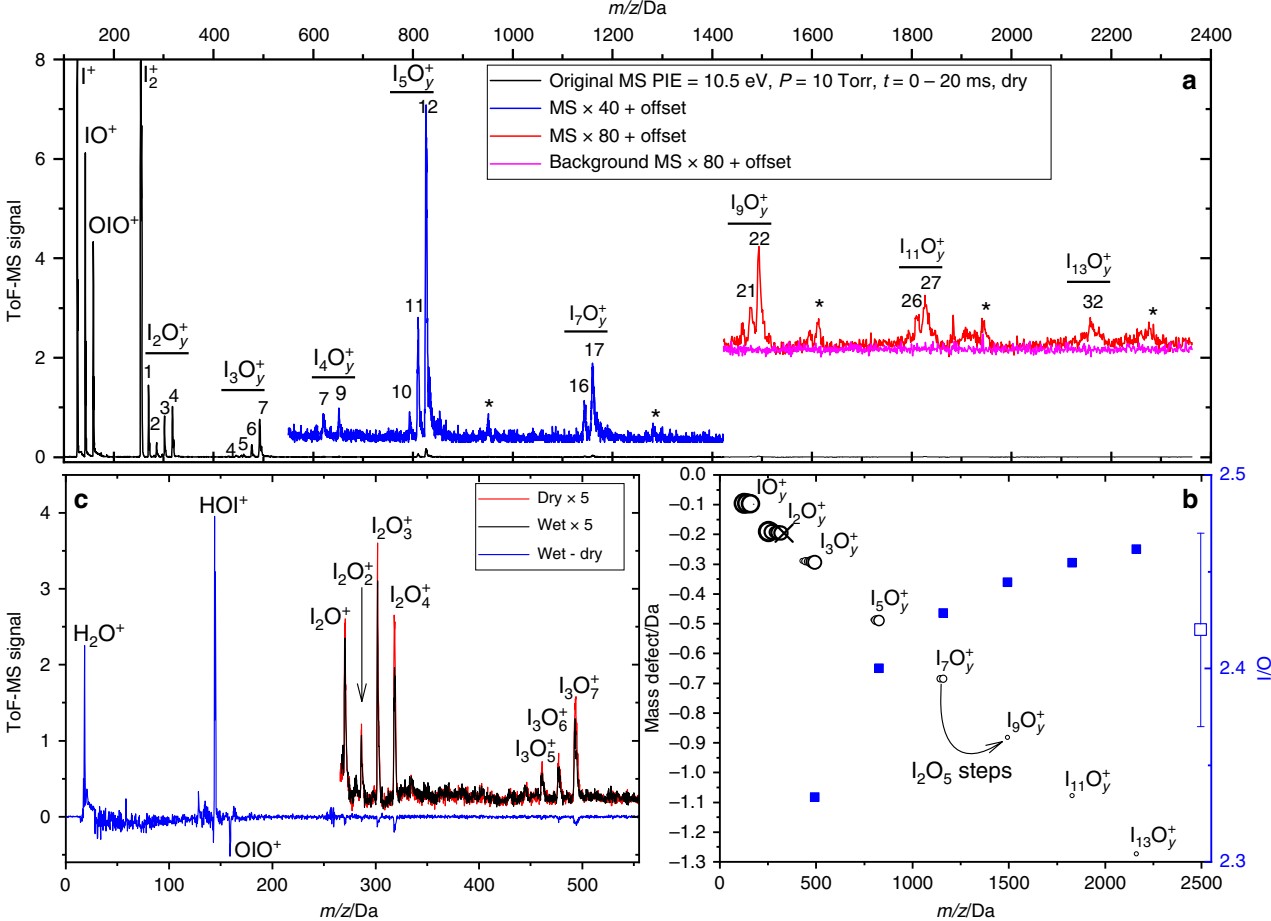

**Fig. 1 Mass spectra and mass defect of iodine oxides. a** Time-of-flight mass spectrum (ToF-MS) at photoionization energy (PIE) of 10.5 eV (10 Torr, [O$_3$] = 2 × 10$^{15}$ molecule cm$^{-3}$, no added H$_2$O) averaged over 20 ms after the photolysis laser pulse. Two sections of the spectrum are displayed with an offset and multiplied by 40 and 100 to show more clearly the peaks corresponding to higher $m/z$ clusters. For the third section (red), the corresponding pre-photolysis background is also shown. The asterisks indicate groups of peaks with even number of iodine atoms. ToF-MS signal in arbitrary units. **b** Mass-defect plot (the difference between molecular mass and the integer mass given by the sum of protons and neutrons in the nuclei) for the main clusters observed. The area of the dots is proportional to the corresponding log-scaled signal intensity. Peak progressions in steps of I$_2$O$_5$ units can be distinguished, but a substantial I$_2$O$_5^+$ signal (x symbol) is absent both at 10.5 eV and 11.6 eV. The blue squares indicate the O/I ratio of the main peaks at the right-hand-side axis (uncertainties are smaller than symbol sizes). The empty square with error bar shows the O/I ratio measured elsewhere for iodine oxide particles (IOPs)[26]. **c** In blue, the difference between spectra with and without water (10.5 eV, 4 Torr, [O$_3$] = 4 × 10$^{14}$ molecule cm$^{-3}$). Negative peaks indicate losses and positive peaks formation of the corresponding species. Sections of the spectra with and without water are shown (black and red lines, respectively), multiplied by 10 for clarity.

water partially inhibits the formation of large clusters, which results in a reduction of the photofragmentation signal.

Observing slow reactions requires increasing the concentration of the excess reactant (H$_2$O) and minimizing the effect of competitive reactions. At 350 Torr and $x$(H$_2$O) = 2%, for PLP-PI delays of up to 5 ms, the peaks observed at low pressure in the mass spectrum remain, and oxyacid clusters do not appear (Fig. 2). The minor peaks at $m/z = 177$ (H$_2$IO$_3^+$) and $m/z = 335$ (HI$_2$O$_5^+$) do not exhibit kinetic change and are decoupled from photolysis-induced chemistry. Thus, they probably result from photofragmentation of large H$_2$O…I$_x$O$_y$ clusters or are produced from deposits around the ToF-MS skimmer.

**Continuous broadband photolysis (BBP) experiments**. We have also performed high-pressure steady-state experiments using continuous BBP. In these experiments, a residence time of a few seconds combined with continuous generation of I atom and high pressure should help in enhancing the concentration of slow-reaction products. Yet again, oxyacid products are not observed

under these conditions (Fig. 2). Upper limits for the rate constants of possible HOIO$_2$-forming reactions can be obtained from kinetic modeling of our BBP experiments, and are discussed below. Separate flow tube measurements of atomic iodine by resonance fluorescence (ROFLEX)[35] were performed to study the removal of I atoms in the presence of MBL-representative concentrations of O$_3$ and H$_2$O at 760 Torr. A slight enhancement of the I atom removal by O$_3$ in the presence of water was observed (Supplementary Fig. 4), which yields an effective rate constant of $k$(I + H$_2$O + O$_3$) = (2.9 ± 1.0) × 10$^{-19}$ cm$^3$ s$^{-1}$ molecule$^{-1}$ for $x$(O$_3$) = 92 ppbv and $k$(I + H$_2$O + O$_3$) = (2.0 ± 0.5) × 10$^{-18}$ cm$^3$ s$^{-1}$ molecule$^{-1}$ for $x$(O$_3$) = 184 ppbv.

Visual inspection of the BBP experiment through a viewport revealed a plume of smoke formed at the I$_2$ inlet under dry conditions (Supplementary Movie 1), which was quenched upon addition of H$_2$O. That is, water appears to hinder nucleation, rather than promoting it by forming HOIO$_2$. The observation of HOIO$_2$ evaporated from particles by resistive heating demonstrates the capability of detecting this molecule by PI-ToF-MS at

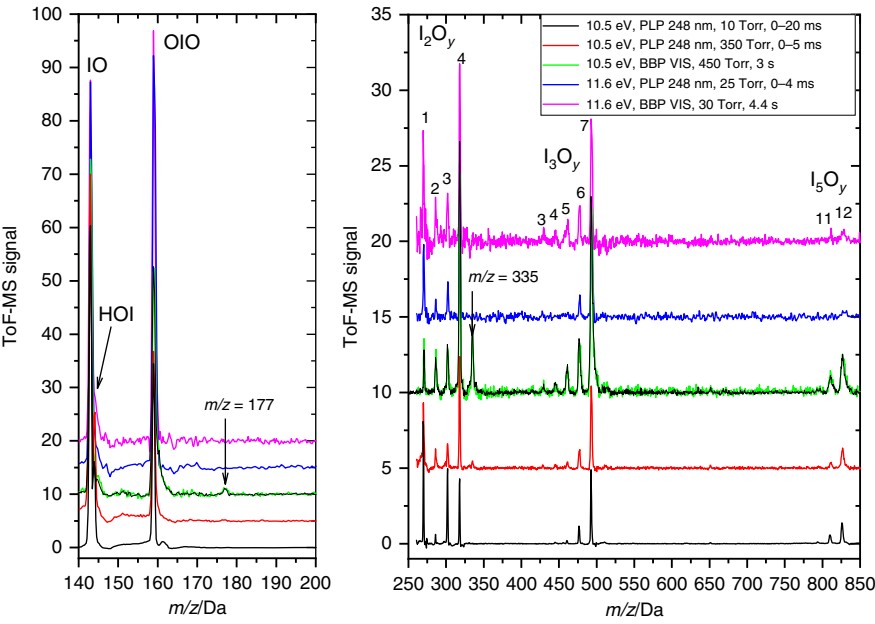

**Fig. 2 Mass Spectra of iodine oxides in the presence of water vapor.** Time-of-flight mass spectra (ToF-MS) acquired under high water mixing ratios (2%) but otherwise different flow tube conditions, photolysis source (PLP pulsed laser photolysis, BBP broadband photolysis), and photoionization energy (PIE). The spectra are displayed in two separated $m/z$ ranges for clarity. Except for the changes in the relative differences between the $I_2O_y$ peaks with pressure and the poorer signal-to-noise ratio at 11.6 eV, the spectra look very similar under all conditions studied. The only instances where some new peaks emerge (red and green lines) are for high pressure (350 and 450 Torr). However, these peaks do not depend on the gas-phase water concentration, as illustrated by the 450 Torr BBP dry spectrum (black line superposed to the green line). These peaks correspond to $m/z = 177$ and $m/z = 355$ and were measured at 10.5 eV, i.e., they can neither be assigned to $HOIO_2^+$ nor $I_2O_5^+$. They were also observed in our previous study under dry conditions[27].

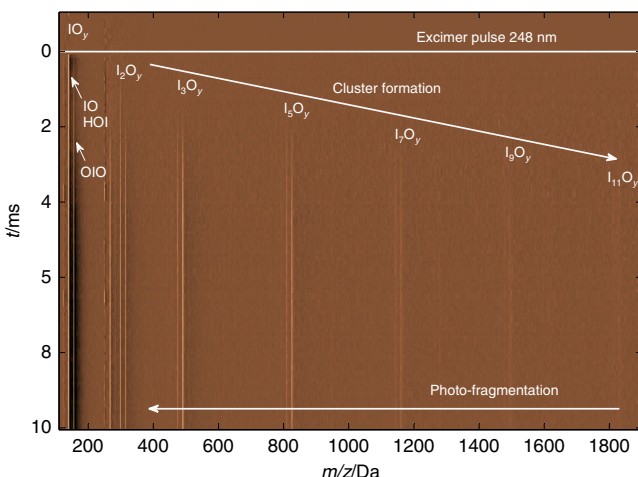

**Fig. 3 Example of time-resolved mass spectrum.** The experiment was started by 248-nm pulsed laser photolysis (PLP) of $O_3$ in the presence of $I_2$ at 10 Torr with 2% water mixing ratio ($6.5 \times 10^{15}$ molecule cm$^{-3}$). The average signal level before PLP has been subtracted. Clear lines indicate time evolution of high signals at certain $m/z$ values (color-coded time-of-flight mass spectrometry (ToF-MS) signal in arbitrary units). Formation of IO ($m/z = 143$), HOI (144) and OIO (159) is followed by the appearance of $I_2O_y$, $I_3O_y$, and larger clusters with the odd number of iodine atoms. At late times all traces grow similarly as a result of photofragmentation of large $I_xO_y$. The time-resolved mass spectra obtained for humid conditions are almost identical to those obtained under dry conditions, except for the presence of HOI and a slight decrease of all $I_xO_y$ signals. In this experiment, $HOIO_2$ would not be expected to be observed due to the photoionization energy (PIE) of 10.5 eV below the ionization threshold.

11.6 eV and provides insights into the formation of particulate $HOIO_2$ (Fig. 5). BBP of $I_2/O_3$ mixtures at room temperature generates $I_xO_y$ and IOPs that travel for a few seconds down the flow tube toward the detection region. A few millimeters upstream of the sampling volume, the carrier gas and the particles are resistively heated (Supplementary Fig. 2) in order to observe evaporation products. The current through the filament was set to a value such that the $I_xO_y$ molecules carried by the gas thermally dissociated in the absence of water (disappearance of the higher mass peaks). Thus, the observed $HOIO_2$ and HOI cannot be products of gas-phase high-temperature $I_xO_y + H_2O$ reactions, but evaporation products of IOPs formed upstream in the presence of water. Under dry conditions, the oxyacid evaporation signals reduce drastically. The other effect of heating IOPs is the expected thermal decomposition of solid/liquid-phase iodine oxides into molecular oxygen and iodine[36,37] (enhancement of the $I_2$ signal).

**Iodine oxide particle-formation mechanism.** The chemistry initiating the formation of iodine oxide clusters and particles is reasonably well known (Supplementary Table 1). Therefore, deviations from the phenomenological growth and decay of IO and OIO documented in previous work[34,38] are considered as a sign of fragmentation, especially if the IO and OIO traces track the growth of larger molecules and clusters. Data at high $IO_x$ concentration clearly suffer from this problem (e.g., experiments #2 and #3 in Supplementary Fig. 3) and cannot be used for kinetic analysis, but it must be pointed out that all datasets are affected to some extent at long delay times. Some of the fragmentation energies of large molecules listed in Supplementary Table 2 are just slightly higher than 10.5 eV or even lower. Two-photon processes may also occur like in the case of water

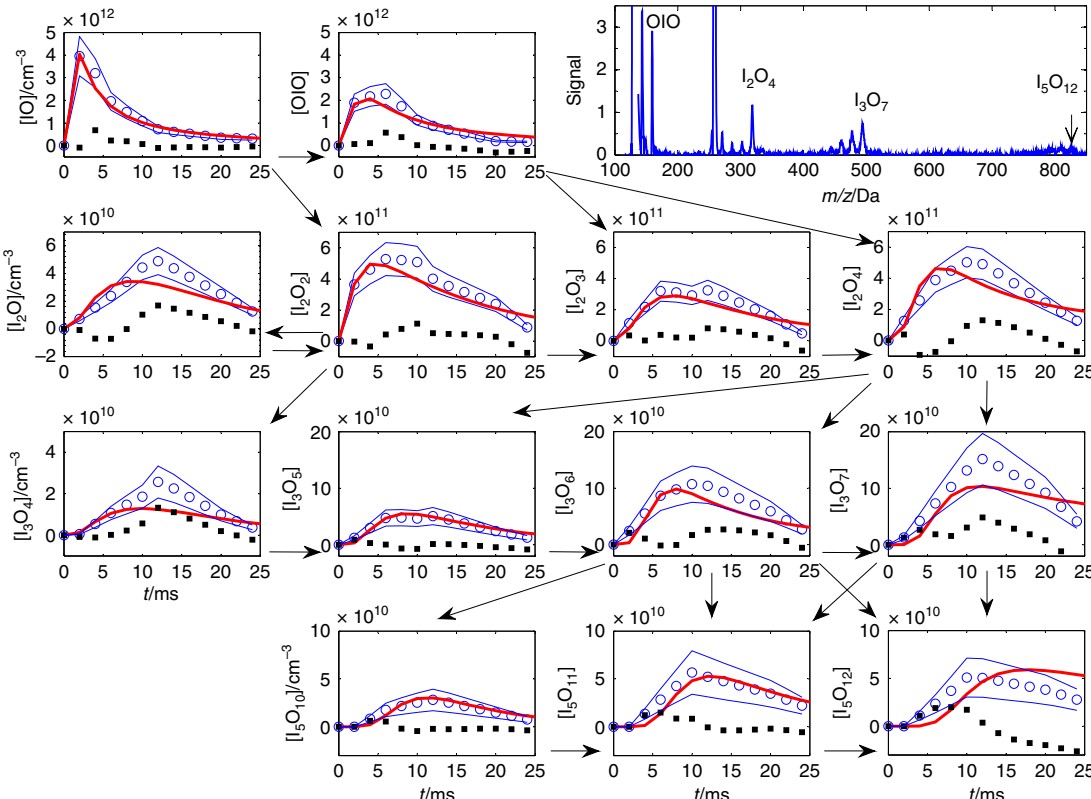

**Fig. 4 $I_xO_y$ kinetics.** Kinetic traces (blue circles, blue lines estimated statistical uncertainty from signal scatter) of the main peaks (blue line in the right upper panel) observed in a pulsed laser photolysis (PLP) experiment initiated by 193-nm photolysis of $I_2$ in the presence of $O_3$ at 9 Torr, with 10.5 eV photoionization (PI). The corresponding mass spectrum averaged for all delays is shown in the upper right corner panel. The data were recorded in the digital mode, and the PLP-PI delays were set manually in the delay generator. All traces (in concentration in $cm^{-3}$ versus time in ms) have been interpolated to a constant step time axis. Note that each row shows $I_xO_y$ with the same number of iodine atoms. The vertical scale is the same for all members of each row, except for the first members of the second and third rows. The red lines show the kinetic traces obtained from a numerical model (Supplementary Tables 1 and 3), and the black squares show the difference between the measurements and the model. The arrows show the main paths assumed in the chemical mechanism (Table 1).

(see "Methods"), although the reason why water is observed by resonance-enhanced multiphoton ionization (REMPI) with an unfocused laser beam is likely related to the high concentrations employed, orders of magnitude higher than the concentrations of $I_xO_y$.

The $I_2O^+$ signal that is consistently detected in all experiments may originate from an $I_2O$ parent neutral rather than from fragmentation, possibly from a bimolecular reaction between IO and the products of its self-reaction. The observation of $I_2O_2^+$ is consistent with IOIO being one of the major products of the IO self-reaction[34,38], although the signal is very low even in the high-pressure experiments, which may indicate some fragmentation to $IO^+$, or a small PI cross section. $I_2O_3^+$ and $I_2O_4^+$ can be linked to the corresponding neutrals, which are known products of the IO + OIO and OIO + OIO reactions. The main cluster progressions observed are separated by $\Delta(m/z) = 334$ ($I_2O_5$), while $I_2O_5$ itself does not appear to form. Thus, $I_2O_2$, $I_2O_3$, and $I_2O_4$ are considered to be the precursors to iodine oxide molecular clusters. However, they do not appear to aggregate stepwise to form clusters. The ion clusters detected have the specific elemental composition $I_{2n+1}O_{5n+m}^+$ ($n \geq 1$, $m = 0, 1, 2$), i.e., odd numbers of iodine atoms. By looking at Supplementary Table 2, it can be seen that while $I_4O_y$ species may fragment to smaller ions, they cannot fragment to $I_3O_y^+$, and therefore we assume that the latter corresponds to $I_3O_y$ molecules formed in the flow tube.

There are many possible reactions that may generate the observed peaks in the PI mass spectra. Thermochemical data are required to eliminate irrelevant reactions from the chemical mechanism (Table 1). Reactions deemed to be endothermic or forming weakly bound adducts based on our calculations are excluded. For example, $I_2O_y$ form generally weak adducts with themselves, and this may explain why the $I_4O_y$ signals are negligible and, as a consequence, clusters with even numbers of iodine atoms do not form[32]. The most stable predicted adduct is $I_4O_8$, formed from the $I_2O_4$ self-reaction. However, our ab initio calculations suggest that this reaction also has an exothermic bimolecular channel making $I_3O_7$. Alternative routes to $I_3O_y$ are $I_2O_y + IO_x$ ($y \neq 3$) followed by ozone-oxidation steps. Ozone has been found not to be required to form IOPs as long as an alternative source of oxygen atoms is available, but its presence leads to enhanced particle formation[26]. Thus, it could help in increasing the O/I ratio within each peak progression if the peaks with lower O/I form first from $I_2O_y + IO_x$.

$I_2O_3$, in particular, does not form stable adducts with other $IO_x$ and $I_2O_y$ (including itself)[32], and we have not found any exothermic bimolecular reaction between $I_2O_3$ and $IO_x$ or $I_2O_y$ (Table 1). However, $I_2O_3$ does not accumulate in the flow tube, as shown in Fig. 4 and Supplementary Fig. 3. Therefore, we assume that it undergoes clustering with larger $I_xO_y$. Further ab initio calculations are required to investigate if this is the actual fate of $I_2O_3$.

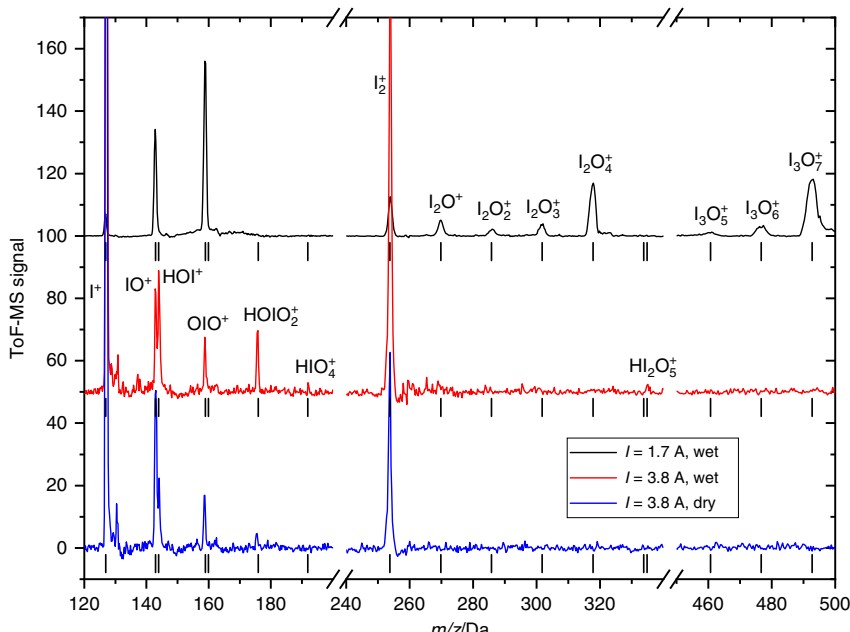

**Fig. 5 Iodine oxide particles (IOP) pyrolysis experiments at 11.6 eV.** The particles are grown along the flow tube, and the carrier gas and particles are passed through a 1-cm diameter resistively heated filament coil just before sampling to detect evaporation products. At a lower filament current of $I = 1.7$ A (lower temperature at the sampling point), the spectrum obtained at 30 Torr with 2% water mixing ratio resembles other broadband photolysis (BBP) spectra in Fig. 2. When the current through the coil is higher (3.8 A), the spectrum changes (red line) as a result of particle evaporation: the higher masses disappear, the $I_2$ signal increases and new masses show up at $m/z = 144$ (HOI$^+$) and $m/z = 176$ (HOIO$_2^+$). When water is not added actively to the flow, the oxyacid ion signals decrease drastically, but still some detectable signals remain due to release of water from the inner walls of the experiment. The vertical black lines indicate the position of the oxide and oxyacids in the m/z axis: from left to right: I, IO, HOI, OIO, HOIO, HOIO$_2$, HIO$_4$, I$_2$, I$_2$O, I$_2$O$_2$, I$_2$O$_3$, I$_2$O$_4$, I$_2$O$_5$, HI$_2$O$_5$, I$_3$O$_5$, I$_3$O$_6$, I$_3$O$_7$. Time-of-flight mass spectrum (ToF-MS) signal in arbitrary units.

The list of reactions added to the initial chemistry in Supplementary Table 1 are listed in Table 2. The rate constants of the selected reactions are then adjusted to model the fast formation and decay of the $I_xO_y$ species, as shown in Fig. 4 (see the section "Kinetic Analysis" in "Methods"). Some reactions are grouped and assumed to have the same rate coefficient. Some rate coefficients are floated by the least-squares algorithm, while some others are changed manually. Not all coefficients can be allowed to float at the same time. Finding a reasonable result implies a few iterations. The criterion is always to minimize the difference between the measured traces and the simulations.

The rapid kinetics and smoke formation are astonishing, considering that the pressure is relatively low and that all reactions are second order. Removal rates of ~50 s$^{-1}$ with concentrations of $10^{11}$ molecule cm$^{-3}$ (Fig. 4 and Supplementary Fig. 3) involve rate constants of the order of $5 \times 10^{-10}$ cm$^3$ s$^{-1}$ molecule$^{-1}$. These are likely to be enabled by the large dipole moments characteristic of iodine oxides[32] inducing long-range capture forces. Kinetic modeling efforts are rendered semi-quantitative at this point by the unknown contribution of fragmentation of large clusters to the smaller molecules. However, a reasonable global fit to all available traces (Fig. 4) may be obtained using standard chemistry (Supplementary Table 1) and the combination of clustering and bimolecular reactions listed in Table 2. In this $I_xO_y$ nucleation mechanism, rapid pressure-independent bimolecular reactions between $I_xO_y$ molecular clusters form larger clusters with the observed odd numbers of iodine atoms (($I_2O_5$)$_n$OIO), e.g.:

$$I_2O_r + I_2O_s \rightarrow I_3O_t + IO_{r+s-t} (r, s = 2 - 4; t = 4 - 7) \quad (1)$$

$$I_3O_u + I_3O_v \rightarrow I_5O_w + IO_{u+v-w} (u, v = 4 - 7; w = 10 - 12). \quad (2)$$

In addition, third-body reactions at their high-pressure limit form aggregates, e.g.:

$$IO_x + I_2O_y \rightarrow I_3O_{x+y} (x = 1, 2; y = 3, 4) \quad (3)$$

$$I_2O_y + I_3O_z \rightarrow I_5O_{y+z} (y = 2 - 4; z = 4 - 7), \quad (4)$$

where the O/I ratio of the aggregates may be subsequently increased through oxidation by O$_3$. HOIO$_2$-containing clusters[23] have not been observed, and therefore no attempt has been made to model them.

**Constraints to iodic acid-formation rates.** Possible sources of gas-phase HOIO and HOIO$_2$ are reactions of IO$_x$ and I$_2$O$_y$ with water listed in Table 1. We have carried out high-level ab initio calculations to explore the transition states toward oxyacid formation from $I_2O_y + H_2O$ ($y = 2$, 3). The gas-phase I$_2$O$_2$ reaction (Supplementary Fig. 5) is more favorable than the I$_2$O$_3$ reaction (Supplementary Fig. 6), while the I$_2$O$_4$ reaction[30] presents a small barrier (Supplementary Fig. 7). We have also hypothesized that wall reactions of iodine oxides with adsorbed H$_2$O may be an additional source of oxyacids in long-residence time experiments and field instruments with long inlets. However, Born–Oppenheimer molecular simulations (BOMD) rule out I$_2$O$_y$ reactions in the air-water interface (see "Methods").

We have selected five exothermic reactions producing oxyacids listed in Table 1 in order to determine their rate constants from the experimental data and from our ab initio calculations. Active addition of high water concentrations in our BBP experiments at a PI energy of 11.6 eV does not result in a detectable signal of HOIO$_2$. Moreover, the co-products of these reactions HOI and HOIO are very weak or absent from the BBP mass spectra obtained at 10.5 eV. This suggests that these reactions are slow

**Table 1 Reaction enthalpies and barriers (in brackets) in kJ mol$^{-1}$ of reactions forming iodine oxides and oxyacids.**

| Reaction | $\Delta H_r$ (298 K), this work[a] | $\Delta H_r$ (298 K), literature[b] |
|---|---|---|
| $IO + IOIO \rightarrow I_2O + OIO$ | −50 | −52 |
| $IO + I_2O_3 \rightarrow I_3O_4$ | | −14 |
| $IO + I_2O_4 \rightarrow I_3O_5$ | | −63 |
| $OIO + IOIO \rightarrow I_2O_3 + IO$ | −51 | −102, −66 |
| $OIO + I_2O_3 \rightarrow I_3O_5$ | | −16 |
| $OIO + I_2O_3 \rightarrow I_2O_4 + IO$ | 55 | 60 |
| $OIO + I_2O_4 \rightarrow I_3O_6$[d] | −83 | −74 |
| $IOIO + IOIO \rightarrow I_2O_3 + I_2O$ | −100 | −154 |
| $IOIO + I_2O_3 \rightarrow I_2O_4 + I_2O$ | 6 | 40 |
| $I_2O_3 + I_2O_3 \rightarrow I_3O_6 + I$ | 99 | |
| $I_2O_3 + I_2O_4 \rightarrow I_3O_6 + IO$ | 43 | |
| $I_2O_3 + I_2O_4 \rightarrow I_3O_7 + I$ | 52 | |
| $I_2O_4 + I_2O_4 \rightarrow I_3O_6 + OIO$ | −12 | |
| $I_2O_4 + I_2O_4 \rightarrow I_3O_7 + IO$[c] | −4 | |
| $I_2O + O_3 \rightarrow IOIO + O_2$ | −137 | |
| $I_2O_2 + O_3 \rightarrow I_2O_3 + O_2$ | −236 | −220 |
| $I_2O_3 + O_3 \rightarrow I_2O_4 + O_2$ | −130 | −95 |
| $I_3O_6 + O_3 \rightarrow I_3O_7 + O_2$ | −177 | |
| $I + H_2O \rightarrow I...H_2O$ | | −20 |
| $I + H_2O + O_3 \rightarrow HOIO_2 + OH$ | −83 | |
| $I + H_2O + O_3 \rightarrow HOIO + HO_2$ | −60 | |
| $IO + H_2O \rightarrow IO...H_2O$ | | −16 |
| $IO + H_2O + O_3 \rightarrow HOIO_2 + HO_2$ | −112 | |
| $OIO + H_2O \rightarrow OIO...H_2O$ | | −27 |
| $OIO + H_2O \rightarrow HOIO + OH$ | | **134** |
| $IOIO + H_2O \rightarrow HOI + HOIO$ | −11 | |
| $I_2O_3 + H_2O \rightarrow I_2O_3...H_2O$ | −38 | −36 |
| $I_2O_3 + H_2O \rightarrow HOI + HOIO_2$ | 1 [32] | |
| $I_2O_4 + H_2O \rightarrow I_2O_4...H_2O$ | | −53, −28 |
| $I_2O_4 + H_2O \rightarrow HOIO...HOIO_2$ | | −104 |
| $I_2O_4 + H_2O \rightarrow HOIO + HOIO_2$ | | 2.9 [3.8] |
| $I_2O_5 + H_2O \rightarrow 2HOIO_2$ | | 5 [**40**], −8 [13] |
| $HOIO + HOIO \rightarrow HOI + HOIO_2$ | 108 [12] | |
| $HOIO + HOIO \rightarrow H_2O + I_2O_3$ | 116 | |
| $HOIO + HOIO_2 \rightarrow HOIO...HOIO_2$ | | −87 |
| $HOIO_2 + HOIO_2 \rightarrow HOIO_2...HOIO_2$ | | −107 |

[a]Normal typescript: B3LYP/6-311 + G(2d,p)(AE). Underlined: CCSD(T)//M06-2X/aug-cc-pVTZ+LANL2DZ.
[b]Calculated from published values of ab initio formation or reaction enthalpies. Normal typescript: CCSD(T)//MP2/aug-cc-pVTZ (AREP) from Galvez et al.[32]; italics: CCSD(T)//B3LYP/aug-cc-pVTZ + AE from Kaltsoyannis and Plane[60]; underlined: CCSD(T)//M06-2X/aug-cc-pVTZ+LANL2DZ from Kumar et al.[30] (concerted pathways); bold: CCSD(T)/CBS//MP2/aug-cc-pVTZ (ECP28) from Khanniche et al.[33].

and cannot compete under our experimental conditions with the reactions removing I$_x$O$_y$ in the "dry" experiments. This, however, does not rule out the possibility that under atmospheric conditions, the situation reverses as a result of the low ambient IO$_x$ ([IO$_x$] <10$^9$ cm$^{-3}$ have been measured in the field, i.e., three to four orders of magnitude lower than in our BBP experiments) and higher water concentrations. Thus, upper limits to the rate constants of these reactions need to be determined as a first step to establish the relative importance of the oxide and oxyacid clusters. This has been done by kinetic modeling of our long-residence time BBP steady-state experiments at 10.5 eV and 11.6 eV PI energy, where each process in Table 3 is included separately in the numerical model with different values of the rate constant. The calculated concentrations of the oxyacid co-products after a time equal to the residence time in the flow tube are used to scale their simulated peaks, which have the typical mass resolution of the measured spectra in the $m/z =$ 140–200 Da range. It is assumed that the PI cross section of each iodine oxyacid is equal to that of the nearest oxide. The upper limit of each rate constant k(I$_x$O$_y$ + H$_2$O) is taken to be that for which the simulated mass spectrometric signal at $m/z = 144$, $m/z = 160$, and/or $m/z = 176$ becomes indistinguishable from the

background noise of the BBP experimental data. These upper limits can then be compared with rate constants calculated with the master equation solver MESMER[39] using the molecular parameters (energies, vibrational frequencies, and rotational constants) derived from our ab initio calculations (Table 3).

A comparison between the simulated ion peaks and the observed mass spectra is shown in Fig. 6. Even though HOIO$_2$ is not expected to be observable at 10.5 eV, the data obtained at this photon energy help in reducing the upper limit of the rate constant of the I$_2$O$_2$ and I$_2$O$_3$ reactions. First, the concentrations of these two oxides, in particular, are enhanced at high pressure (the IO + OIO and OIO + OIO rate constants reach their high-pressure limit)[34]. Second, the water concentration is one order of magnitude higher as a result of the higher pressure (the S/N ratio at 10.6 eV was not high enough to perform experiments above 30 Torr). The upper limits are listed in Table 3. For the I$_2$O$_2$ + H$_2$O and I$_2$O$_4$ + H$_2$O reactions, the rate constant cannot exceed, respectively, 1 × 10$^{-19}$ cm$^3$ s$^{-1}$ molecule$^{-1}$ and 5 × 10$^{-19}$ cm$^3$ s$^{-1}$ molecule$^{-1}$. The upper limit of k(I$_2$O$_3$ + H$_2$O) is 5 × 10$^{-18}$ cm$^3$ s$^{-1}$ molecule$^{-1}$, but owing to the large barrier of the I$_2$O$_3$ + H$_2$O reaction (Supplementary Fig. 6), its rate constant is expected to be at least one order of magnitude lower, as indicated by the master equation calculation

**Table 2 Tentative mechanism of iodine oxide cluster formation (initiated by the reactions included in Supplementary Table 1)[a].**

| # | Reaction[b] | $k$/cm$^3$ s$^{-1}$ molecule$^{-1c}$ | Grouped reactions | Fit |
|---|---|---|---|---|
| 1 | $IO + IOIO \rightarrow I_2O + OIO$ | $5 \times 10^{-12}$ | | Floated |
| 2 | $OIO + IOIO \rightarrow I_2O_3 + IO$ | $1 \times 10^{-11}$ | | Floated |
| 3 | $IOIO + IOIO \rightarrow I_2O_3 + I_2O$ | $1 \times 10^{-11}$ | =2 | |
| 4 | $I_2O + O_3 \rightarrow IOIO + O_2$ | $8 \times 10^{-14}$ | | Manual |
| 5 | $I_2O_2 + O_3 \rightarrow I_2O_3 + O_2$ | $4 \times 10^{-14}$ | | Manual |
| 6 | $I_2O_3 + O_3 \rightarrow I_2O_4 + O_2$ | $8 \times 10^{-14}$ | | Manual |
| 7 | $OIO + IOIO \rightarrow I_3O_4$ | $3 \times 10^{-12}$ | | Floated |
| 8 | $IO + I_2O_4 \rightarrow I_3O_5$ | $3 \times 10^{-11}$ | | Floated |
| 9 | $OIO + I_2O_4 \rightarrow I_3O_6$ | $5 \times 10^{-11}$ | | Floated |
| 10 | $I_2O_4 + I_2O_4 \rightarrow I_3O_6 + OIO$ | $5 \times 10^{-12}$ | | Floated |
| 11 | $I_2O_4 + I_2O_4 \rightarrow I_3O_7 + IO$ | $1 \times 10^{-10}$ | | Floated |
| 12 | $I_3O_4 + O_3 \rightarrow I_3O_5 + O_2$ | $8 \times 10^{-14}$ | | Manual |
| 13 | $I_3O_5 + O_3 \rightarrow I_3O_6 + O_2$ | $1 \times 10^{-13}$ | | Manual |
| 14 | $I_3O_5 + O_3 \rightarrow I_3O_7 + O_2$ | $1 \times 10^{-13}$ | | Manual |
| 15 | $I_3O_6 + I_2O_3 \rightarrow I_5O_9$ | $1.7 \times 10^{-10}$ | | Floated |
| 16 | $I_3O_7 + I_2O_3 \rightarrow I_5O_{10}$ | $1.7 \times 10^{-10}$ | =15 | |
| 17 | $I_3O_5 + I_2O_4 \rightarrow I_5O_9$ | $1.7 \times 10^{-10}$ | =15 | |
| 18 | $I_3O_6 + I_2O_4 \rightarrow I_5O_{10}$ | $1.7 \times 10^{-10}$ | =15 | |
| 19 | $I_3O_7 + I_2O_4 \rightarrow I_5O_{11}$ | $1.9 \times 10^{-10}$ | | Floated |
| 20 | $I_3O_6 + I_3O_6 \rightarrow I_5O_{11} + IO$ | $4 \times 10^{-10}$ | | Floated |
| 21 | $I_3O_6 + I_3O_7 \rightarrow I_5O_{11} + OIO$ | $4 \times 10^{-10}$ | =21 | |
| 22 | $I_3O_6 + I_3O_7 \rightarrow I_5O_{12} + IO$ | $2 \times 10^{-10}$ | | Floated |
| 23 | $I_3O_7 + I_3O_7 \rightarrow I_5O_{12} + OIO$ | $2 \times 10^{-10}$ | =22 | |
| 24 | $I_5O_9 + O_3 \rightarrow I_5O_{10} + O_2$ | $3 \times 10^{-13}$ | | Manual |
| 25 | $I_5O_{10} + O_3 \rightarrow I_5O_{11} + O_2$ | $3 \times 10^{-13}$ | | Manual |
| 26 | $I_5O_{11} + O_3 \rightarrow I_5O_{12} + O_2$ | $2 \times 10^{-13}$ | | Manual |
| 27 | $I_5O_9 + I_2O_3 \rightarrow I_7O_{12}$ | $1.7 \times 10^{-10}$ | =15 | |
| 28 | $I_5O_{10} + I_2O_3 \rightarrow I_7O_{13}$ | $1.7 \times 10^{-10}$ | =15 | |
| 29 | $I_5O_{11} + I_2O_3 \rightarrow I_7O_{14}$ | $1.7 \times 10^{-10}$ | =15 | |
| 30 | $I_5O_{12} + I_2O_3 \rightarrow I_7O_{15}$ | $1.7 \times 10^{-10}$ | =15 | |
| 31 | $I_5O_9 + I_2O_4 \rightarrow I_7O_{13}$ | $1.9 \times 10^{-10}$ | =19 | |
| 32 | $I_5O_{10} + I_2O_4 \rightarrow I_7O_{14}$ | $1.9 \times 10^{-10}$ | =19 | |
| 33 | $I_5O_{11} + I_2O_4 \rightarrow I_7O_{15}$ | $1.9 \times 10^{-10}$ | =19 | |
| 34 | $I_5O_{12} + I_2O_4 \rightarrow I_7O_{16}$ | $1.9 \times 10^{-10}$ | =19 | |
| 35 | $I_5O_{12} + I_3O_6 \rightarrow I_7O_{17} + IO$ | $1 \times 10^{-10}$ | | Floated |
| 36 | $I_5O_{12} + I_3O_7 \rightarrow I_7O_{17} + OIO$ | $3 \times 10^{-10}$ | | Floated |
| 37 | $I_5O_{11} + I_5O_{11} \rightarrow IOP$ | $6 \times 10^{-10}$ | | Floated |
| 38 | $I_5O_{11} + I_5O_{12} \rightarrow IOP$ | $6 \times 10^{-10}$ | =38 | |
| 39 | $I_5O_{12} + I_5O_{12} \rightarrow IOP$ | $7 \times 10^{-10}$ | | Floated |

[a]Ab initio thermochemistry is available up to $I_4O_y$. Endothermic reactions and reactions forming weakly bound complexes are not included (see Table 1). For larger clusters, reactions are included in order to explain observed peaks and relative peak intensities.
[b]For simplicity, only the forward association reactions are considered, i.e., unimolecular decomposition of the adduct is ignored, and therefore these are effective rate constants.
[c]Rate constants from the global fit in Fig. 2.

included in Table 3. The $I_2O_2 + H_2O$ reaction and the $I_2O_4 + H_2O$ reaction (concerted path[30]) present, respectively, a submerged barrier of $-7.5$ kJ mol$^{-1}$ (Supplementary Fig. 5) and a low barrier of 3.4 kJ mol$^{-1}$ (Supplementary Fig. 7) at the CCSD(T)//M06-2X/ aug-cc-pVDZ+LANL2DZ level of theory, which yields calculated rate constants higher than the estimated upper limits deduced from the experimental data.

The upper limit determined for the rate constant of a hypothetical composite reaction where atomic iodine complexes with water, and the resulting $H_2O\ldots I$ adduct reacts with $O_3$ to form oxyacids (Fig. 6a, e) can be compared with the rate constants obtained from the resonance fluorescence experiments on the removal of atomic iodine by $O_3$ in the presence of water vapor. The enhanced removal of I by $O_3$ in the presence of $H_2O$ observed with the ROFLEX for $x(O_3) = 92$ ppbv ($k(I + H_2O + O_3) = (2.9 \pm 1.0) \times 10^{-19}$ cm$^3$ s$^{-1}$ molecule$^{-1}$) appears to agree with the upper limit determined from the ToF-MS experiments: $k(I + H_2O + O_3) < 2.8 \times 10^{-19}$ cm$^3$ s$^{-1}$ molecule$^{-1}$. However, this is coincidental. The $O_3$ concentration in the ToF-MS

experiments was three orders of magnitude larger than in the ROFLEX experiments, indicating that the rate constant in the former should be, if linearly extrapolated from the ROFLEX result, of the order of $2.9 \times 10^{-16}$ cm$^3$ s$^{-1}$ molecule$^{-1}$, i.e., well above the detection limit of $\sim 10^{-18}$ cm$^3$ s$^{-1}$ molecule$^{-1}$. Thus, the yield of HOIO and $HOIO_2$ from this process is <0.001, implying an oxyacid-formation rate constant of the order of $10^{-22}$ cm$^3$ s$^{-1}$ molecule$^{-1}$ for atmospherically relevant $O_3$ concentrations. Therefore, the composite $I + H_2O + O_3$ reaction is not a sufficiently fast source of $HOIO_2$. This is very important because, as discussed in Supplementary Note 1, oxyacids can only be effective IOP nucleating species if their formation is decoupled from $I_xO_y$: otherwise, oxyacid nucleation needs to proceed through two slow second-order processes to form the first cluster. To explore the relative contribution of iodine oxides and oxyacids to the nucleation of IOPs, we have modeled a BBP flow tube experiment with longer residence time (200 s). Calculated rate constants of oxyacid–oxyacid and oxide–oxyacid aggregation and dissociation rates are listed in Supplementary Table 4. Modeled

**Table 3 Calculated rate constants of possible oxyacid sources ($T = 295$ K).**

| Reaction | $k$ MESMER | | $k$ upper-limit experimental[e] |
|---|---|---|---|
| | **30 Torr** | **760 Torr** | |
| $I + H_2O \rightarrow H_2O...I$ | $2.4 \times 10^{-13}$ cm$^3$ s$^{-1}$ | $4.7 \times 10^{-12}$ cm$^3$ s$^{-1}$ | |
| $H_2O...I \rightarrow I + H_2O$ | $1.7 \times 10^6$ s$^{-1}$ | $3.4 \times 10^7$ s$^{-1}$ | |
| $\underline{H_2O...I + O_3 \rightarrow HOIO/HOIO_2 + HO_2/OH}$ | $100$ s$^{-1}$ | $100$ s$^{-1a}$ | |
| $I + H_2O\ (+ O_3) \rightarrow HOIO/HOIO_2 + HO_2/OH$ | $2.8 \times 10^{-19}$ cm$^3$ s$^{-1}$ | $2.1 \times 10^{-19}$ cm$^3$ s$^{-1}$ | $2.8 \times 10^{-19}$ cm$^3$ s$^{-1}$ (**a, e**) |
| $IO + H_2O \rightarrow H_2O...IO$ | $1.9 \times 10^{-15}$ cm$^3$ s$^{-1}$ | $4.6 \times 10^{-14}$ cm$^3$ s$^{-1}$ | |
| $H_2O...IO \rightarrow IO + H_2O$ | $1.6 \times 10^7$ s$^{-1}$ | $4.0 \times 10^8$ s$^{-1}$ | |
| $\underline{H_2O...IO + O_3 \rightarrow HOIO_2 + HO_2}$ | $100$ s$^{-1}$ | $100$ s$^{-1b}$ | |
| $IO + H_2O\ (+ O_3) \rightarrow HOIO_2 + HO_2$ | $2.0 \times 10^{-21}$ cm$^3$ s$^{-1}$ | $2.0 \times 10^{-21}$ cm$^3$ s$^{-1}$ | $2.0 \times 10^{-20}$ cm$^3$ s$^{-1}$ |
| $I_2O_2 + H_2O \rightarrow HOI + HOIO$ | | | $1 \times 10^{-19}$ cm$^3$ s$^{-1}$ (**b**) |
| $I_2O_3 + H_2O \rightarrow HOI + HOIO_2$ | $2.3 \times 10^{-20}$ cm$^3$ s$^{-1}$ | | $2 \times 10^{-18}$ cm$^3$ s$^{-1}$ (**c**) |
| | Eckart tunneling$^c$: $3.8 \times 10^{-19}$ cm$^3$ s$^{-1}$ | | |
| $I_2O_4 + H_2O \rightarrow HOIO + HOIO_2$ | $9.3 \times 10^{-16}$ cm$^3$ s$^{-1}$ | $8.0 \times 10^{-16}$ cm$^3$ s$^{-1}$ | $5 \times 10^{-19}$ cm$^3$ s$^{-1}$ (**d**) |
| $\rightarrow HOIO...HOIO_2{}^d$ | $0.2 \times 10^{-16}$ cm$^3$ s$^{-1}$ | $1.4 \times 10^{-16}$ cm$^3$ s$^{-1}$ | |

$^a$Adjusted to obtain a net rate constant equal to the effective rate constant determined by resonance fluorescence with the ROFLEX machine. For $[O_3] = 2.5 \times 10^{12}$ molecule cm$^{-3}$ (100 ppbv at 760 Torr), the corresponding rate constant would need to be: $k = 4 \times 10^{-11}$ cm$^3$ s$^{-1}$ molecule$^{-1}$.
$^b$The same loss rate of the I-water adduct is assumed for the IO-water adduct.
$^c$Quantum-mechanical tunneling is incorporated in the master equation using a parabolic Eckart-type barrier[39,61].
$^d$The HOIO...HOIO$_2$ potential well is deep enough to enable collisional stabilization of the adduct at high pressure. We have no evidence of a peak at $m/z = 336$ in our high-pressure mass spectra.
$^e$Bold letters in brackets refer to the panel of Fig. 6 in the main text showing the simulation from which the upper limit is obtained. For $I_2O_y + H_2O$, the upper limit is better constrained by the experiment at 450 Torr with 10.5 eV PI photon energy (thee co-products HOI and HOIO have ionization energies below 10.5 eV).

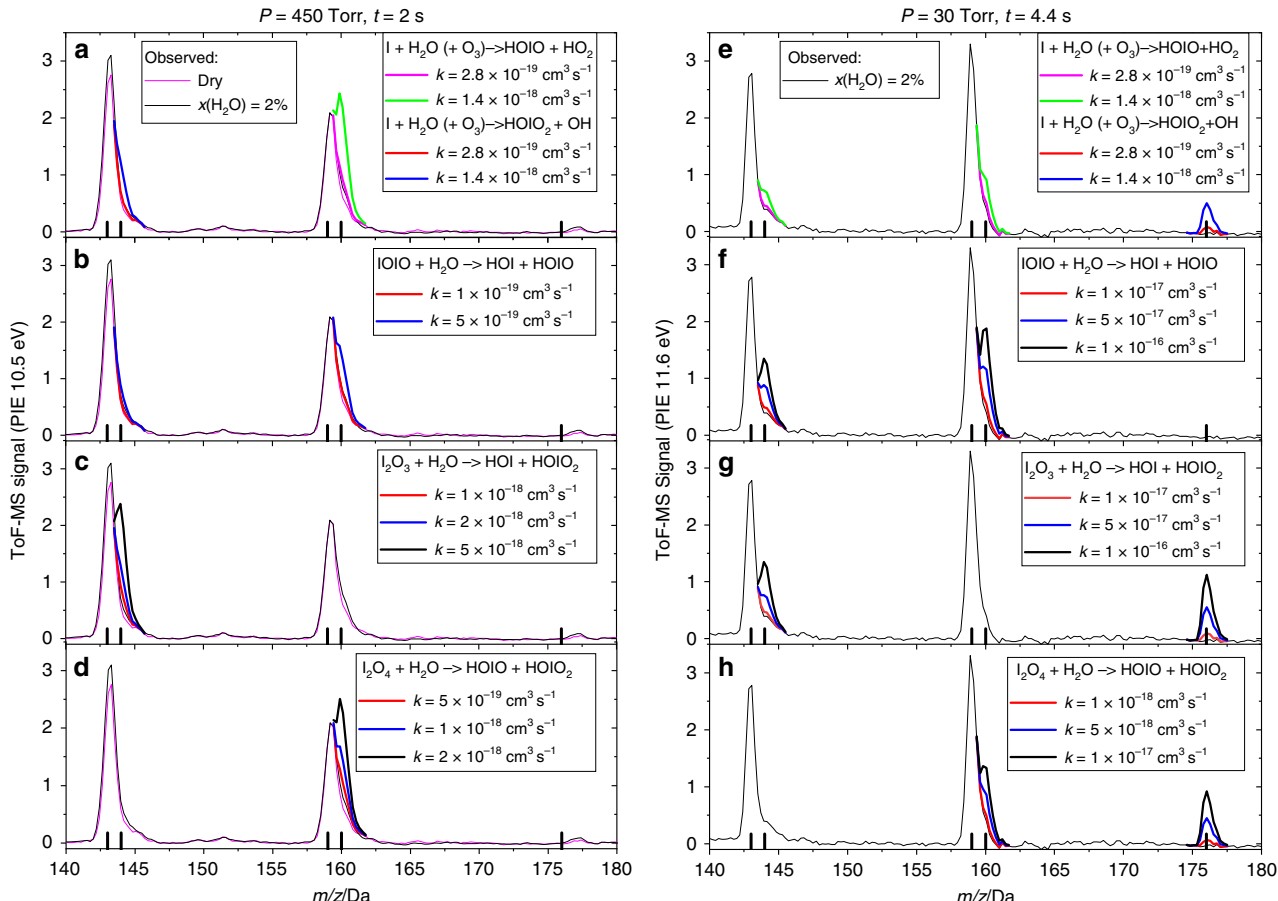

**Fig. 6 Determination of upper limits to oxyacid-formation rate constants.** Time-of-flight mass spectra (ToF-MS) obtained with photoionization energies (PIE) of 10.5 eV (panels **a–d**) and 11.6 eV (panels **e–h**) in broadband photolysis (BBP) experiments at 450 Torr and 30 Torr, respectively, without added water (blue line) and with 2% water mixing ratio (black line). Note that the labels of $y$ and $x$ axes are the same for all panels in the same column. The vertical black lines indicate the position of the oxide and oxyacids in the $m/z$ axis: from left to right: IO, HOI, OIO, HOIO, HOIO$_2$. Simulated oxyacid ion peaks (thick lines) resulting from four different kinetic modeling scenarios with different rate constants are overlaid to the measured spectra: $I + H_2O + O_3 \rightarrow HOIO/HOIO_2 + HO_2/OH$ (**a, b**) $IOIO + H_2O \rightarrow HOIO_2 + HO_2$ (**b, f**), $I_2O_3 + H_2O \rightarrow HOI + HOIO_2$ (**c, g**), $I_2O_4 + H_2O \rightarrow HOIO + HOIO_2$ (**d, h**), and $I + H_2O + O_3 \rightarrow OH + HOIO_2$. Note that reactions producing HO$_x$ also would result in the generation of HOI (panels **a, e**).

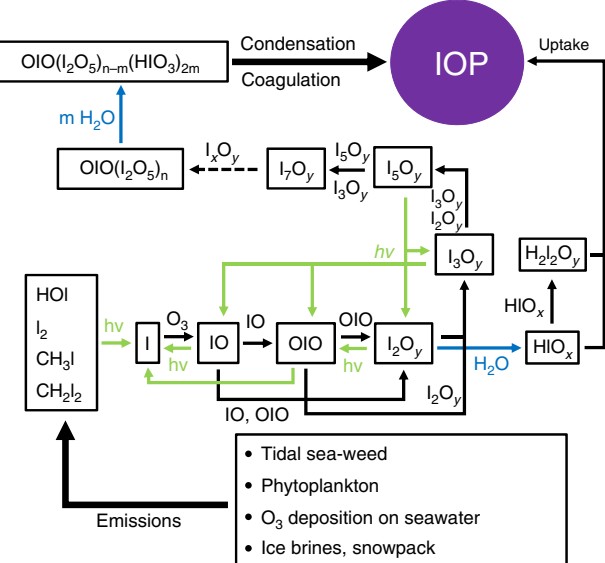

**Fig. 7 Proposed mechanism of iodine oxide new particle formation.**
Iodine-bearing molecules are emitted from the ocean surface and marine biota[4,5,64,65] and photo-oxidized in the presence of $O_3$, leading to the formation of OIO[34] and $I_2O_y$ molecules[27]. According to the findings of this work, $I_2O_y$ reactions lead to the formation of molecular clusters with composition $OIO(I_2O_5)_n$, which transition into $HOIO_2$-containing clusters and iodine oxide particles (IOP) by progressive hydration (the derivation of the mechanism is explained in the "Results" section). Further uptake of water, organic acids, and $H_2SO_4$[26] leads to particle formation. The formation of iodic acid ($HOIO_2$) by the reaction of $H_2O$ with $I_2O_y$ cannot be disproved, but appears to be too slow for $HOIO_2$ nucleation to occur. It is likely that $HOIO_2$ formed in this way—or from the OIO + OH reaction—is taken up by iodine oxide clusters and aerosol.

concentrations are presented in Supplementary Fig. 8 as a function of water vapor concentration ([$H_2O$] concentrations between laboratory experiments[23] and ambient). Despite the many assumptions involved in this exercise, a conclusion that can be drawn is that molecular clusters descending directly from HOIO and $HOIO_2$ can only become dominant at high water concentration ($>5 \times 10^{16}$ molecule cm$^{-3}$), and if the direct precursor of these oxyacids is atomic iodine. If the precursors of these oxyacids are oxides—which is more likely—nucleation is started by iodine oxides.

## Discussion

Our laboratory results strongly support an atmospheric gas-to-particle conversion mechanism where the initial clustering steps are driven by $I_xO_y$ bimolecular and third-body reactions, both under dry and wet conditions. The mechanism of atmospheric IOP formation proposed in Fig. 7 is based on the set of reactions (Table 2) that best reproduces the time-resolved mass spectrometric data (Fig. 4) in the presence and absence of water, as described in the "Results" section. The IOP pyrolysis experiments show that HOI and $HOIO_2$ are constituents of these particles, even though we did not observe them in the gas phase at room temperature, which implies that they are not indispensable IOP gas-phase precursors. This evidences the transformation of the $I_xO_y$ clusters into oxyacid-containing clusters within the flow tube residence time, which confirms earlier predictions about the composition of $I_2O_5$ IOPs in the MBL[28]. This scenario is represented in Fig. 7 by the hydration of the $I_2O_5$ units present in the iodine oxide clusters with the

composition $(I_2O_5)_n$OIO. We reported previously that the number and the size of IOPs reduces when they are formed in the presence of water[26]. Particle capillary collapse following water uptake could not alone explain those observations. The attachment of water molecules to molecular clusters may have the effect of reducing cluster formation and particle coagulation rates. At the molecular cluster level of the mass spectrometric observations discussed here, the addition of water results in a consistent reduction of the intensity of all masses without concomitant formation of products, which we have interpreted as inhibition of photofragmentation of larger clusters. We found previously[26,32] that the key iodine oxide $I_2O_4$ forms a relatively stable complex with water, such that under the conditions prevailing in the MBL, a large fraction of $I_2O_4$ molecules would be hydrated as $I_2O_4$…$H_2O$ (> 40%). This could block the $I_2O_4$ halogen-bonding sites, and the subsequent necessary geometrical rearrangement for the reaction of $I_2O_4$ with itself or with $I_xO_y$ to proceed may result in barriers in the corresponding PES. The same may occur for larger molecular clusters. It should be noted that $I_xO_y$…$H_2O$ clusters are not observed in our experiments, which may be because they promptly dissociate following absorption of vacuum ultraviolet (VUV) photons.

A possible role of iodine in the formation and growth of new particles near the tropical tropopause has been proposed[9]. This would be favored by low water concentrations, but it must be noted that if the mixing ratios of $I_xO_y$ are low, the clustering process slows down compared to photolysis (photolysis lifetimes of tens of seconds[10,40] versus reaction lifetimes of hours, Supplementary Table 5). In mid-latitude coastal regions, particle production probability shows a strong negative correlation with water vapor fluxes and relative humidity[31]. This is consistent with our laboratory observations of water slowing down IOP formation[26], and not consistent with a mechanism which results in a positive dependence on water vapor. In the polar MBL, active iodine ($IO_x$) mixing ratios are in general lower than in macroalgae-rich mid-latitude locations[5], and consequently lower $I_xO_y$ mixing ratios should result in slower IOP formation rates (Supplementary Table 5). However, nucleation is likely to be favored not only by enhanced stability of molecular clusters at low temperature[26] but also because the water content of the atmosphere is lower. In fact, nucleation events observed in the Arctic[21] are coincident with periods of low RH.

Although we cannot rule out the formation of gas-phase $HOIO_2$ (and HOIO) from $I_2O_y$ in the atmosphere (see lifetimes in Supplementary Table 5), we have shown that oxyacids are unlikely to be major nucleating species of IOPs. Moreover, we have evidence indicating that CI-API-ToF-MS $IO_3^-$ signals attributed to gas-phase $HOIO_2$ prior to reaction using two reagent ions—nitrate and acetate—may require some re-interpretation (see Supplementary Note 2 and Supplementary Table 3). Besides collisional fragmentation issues of this technique[41], $IO_3^-$ ions are exothermic reaction products of the dissociative charge transfer between $I_2O_y$ ($y = 2$, 3, 4) and $NO_3^-$ (and $CH_3COO^-$), and therefore it is likely that $I_2O_y$ contributes significantly to the $IO_3^-$ signal. The apparent paradox is that the $IO_3^-$ CI-API-ToF measurements[23] remains a very useful proxy for tracking iodine nucleation in the field, because it may result from ionization of $I_2O_y$ and/or $HOIO_2$ generated by $I_2O_y + H_2O$. Nevertheless, connecting the dots between these observations and those of precursors and smaller gas-phase species ($I_2$, I, IO, HOI, OIO)[5] is crucial to facilitating the development of a chemical mechanism that can be adapted for atmospheric modeling purposes. To our best knowledge, ours is the first complete iodine gas-to-particle conversion mechanism (Fig. 7) supported by direct experimental evidence.

Future modeling work should adapt this scheme to carry out a global assessment of the contribution of iodine new particle formation to the atmospheric aerosol burden and its associated radiative forcing.

## Methods

**Pulsed laser and continuous broad band photolysis.** A PLP system (Supplementary Fig. 1) was employed to generate iodine oxides from the 193-nm or 248-nm photolysis of $I_2/O_3$ mixtures in a 3.75-cm diameter tubular reaction cell. Alternatively, a Xenon lamp was used for continuous BBP of $I_2$ in the presence of $O_3$. A flow of 0.2–20 slm of $N_2$ carrier gas (99.999%) was introduced in the reactor, depending on the target pressure. $I_2$ molecules were entrained in the carrier flow by passing a smaller flow of carrier gas (10–500 sccm) through a heated, temperature-controlled flask containing $I_2$ crystals. $O_3$ was added by flowing $O_2$ (10–200 sccm) through an electrical discharge. The excimer laser and the lamp beams were passed unfocussed through a quartz viewport along the tube main axis. An atmospheric pressure bubbler was employed to entrain water vapor into the carrier flow at approximately the equilibrium vapor pressure, resulting in a water mixing ratio of ~2% in the flow tube ($[H_2O] = 2.3 \times 10^{17}$ molecule $cm^{-3}$ at 350 torr and 295 K). The pressure in the reactor was set by a throttle valve placed between the reactor and an Edwards 80 roots blower—oil rotary pump combination. The flows were set using MKS calibrated mass flow controllers, and pressure was measured using a set of 10 Torr and 1000 Torr MKS Baratron pressure transducers. $O_3$ and $I_2$ concentrations were measured using a Herriott-type absorption cell situated upstream of the reactor[42].

**Photoionization time-of-flight mass spectrometry.** PI-ToF-Ms systems at Leeds University have been employed extensively to study reaction kinetics relevant to combustion-related and tropospheric organic and halogen chemistry[43–45], and upper atmospheric chemistry of meteoric metals[46]. The first direct observation of $I_xO_y$ ($x \geq 2$) gas-phase species was achieved using the older of these two systems[27]. The newer machine employed in this work has been described in detail in a previous publication[46].

The gas in the reactor tube (Supplementary Fig. 1) is sampled on the axis via a skimmer cone with a 200-μm pinhole using a scroll pump-backed turbo-molecular pump. PI of molecules and clusters takes place in the ionization chamber ($P = 10^{-4}$ –$10^{-5}$ Torr) before mass spectrometric detection. Alternatively, a differentially pumped roughing chamber can be inserted between the reactor and the PI chamber for experiments requiring high pressure ($P > 20$ Torr). In this configuration, the gas is sampled via a 500-μm pinhole into the roughing chamber, forming a high-density jet that is directed toward the 200-μm pinhole skimmer cone. The PI chamber is fitted with viewports, allowing a pulsed laser beam to be directed through the high-density region of the sampled gas jet. VUV ionization is achieved by tightly focusing in a rare-gas cell (Xe) the third harmonic of a Nd:YAG laser (Continuum Surelite 10-II), which produces VUV radiation by frequency tripling (118.2 nm, or equivalently 10.5 eV)[47,48]. The 118.2-nm radiation is delivered at the PI chamber by a waveguide, which is a ¼" glass tube with either constant inner diameter or tapered for enhanced PI efficiency. This technique shows an excellent detection limit for organic molecules such as acetone[43] ($10^{11}$ molecule $cm^{-3}$ in a single accumulation in analog mode), which is used as a standard for optimizing the performance of the instrument. The conversion efficiency of frequency tripling in a rare gas is low (~$10^{-5}$), and therefore VUV PI is a viable method for molecules with substantial PI cross-sections (>$10^{-18}$ $cm^2$ molecule$^{-1}$). Possible interference from the 355-nm fundamental, if not separated from the VUV third harmonic, needs to be considered. This is important for molecules with UV–vis absorption bands, low ionization energy and high two-photon PI cross-sections. For example, water is detected at $m/z = 18$ (Fig. 1) using the 118.2-nm (10.5 eV) PI beam, even though its ionization potential is 12.621 eV. This most likely results from REMPI, where the intermediate state at 118.2 nm is the C electronic state or a Rydberg state of water[49], and a 355-nm photon is enough for reaching the ground state of the ion. Although the PI beam is diverging in the ionization volume, this appears to be compensated by the high water concentrations employed. A 106.7 nm (11.6 eV) PI beam can also be generated by frequency tripling the second harmonic of a 532-nm pumped dye laser running on DCM dye. The power of this beam is approximately an order of magnitude lower than that of the 10.5 eV beam.

The positive ions resulting from PI of molecules in the sampled gas jet are accelerated toward the ToF-MS (Kore Technology) by a set of ion optics. This machine has been described in detail elsewhere[46]. A pre-amplifier coupled to the positive ion detector provides simultaneous analog and digital outputs. The analog output, which can be modulated linearly by varying the detector gain, is intended for registering large signals from species with high PI efficiency and/or high concentration. This output is recorded by using a digital oscilloscope (Picoscope 6000). The mass peaks appear in the time axis of the oscilloscope delayed with respect to the PI laser pulse by a characteristic time of flight. The peaks are then gate-integrated and passed on to a PC for further analysis. A limitation of this method is that strong signals cause detector overload after the corresponding peak during a significant lapse, which appears as a negative signal with respect to the spectrum baseline. Thus, peaks of other species arriving at the detector closely after

the species causing a large peak are affected by this interference. Overloads can be eliminated by tuning down the detector gain, but this also reduces the sensitivity to small signals. To prevent detector overload by $I^+$ and $I_2^+$, their mass peaks were gated out by pulsed biasing of the ion optics using a house-built dual-gating box.

The digital output is provided to a counting system (time-to-digital convertor, TDC) by means of a fast comparator. The Kore Technology TDC counts ions within a sequence of flight-time bins (0.5 ns) referred to the PI laser pulse. In this way, a histogram of counts versus time of flight, (i.e., a mass spectrum) is built. The counting method is suitable for capturing small signals (low concentration or low PI efficiency). The signal remains proportional to concentration for low counting rates. Although spectra measured with this technique are not affected by detector overload, for high counting rates, departure from linearity occurs and the peaks appear saturated.

Lasers and detectors are synchronized using a delay generator (Quantum Composers, 9518). In every measuring cycle, the delay generator triggers the Nd: YAG laser and after a few microseconds triggers the TDC or the scope to establish the time zero event of each measuring cycle. In addition, the delay generator triggers the excimer laser that initiates chemistry by photolyzing a radical precursor. The delay between the photolysis and the PI laser is scanned in order to sample reaction kinetics. In a typical digital experiment, 1000 digital spectra were accumulated for every reactant concentration and laser pulse delay, which at a 5-Hz laser-repetition rate takes 200 s. The built-in software of the MS-ToF does not handle synchronization of the delay generator and the TDC, and therefore the delays were set manually, at the cost of a poorer time resolution. Once the accumulation was complete, the delay between laser pulses and/or concentration of reagents was changed, and a new accumulation started. Kinetic experiments using the digital output are shown in Fig. 4 and Supplementary Fig. 3 (Exp. #1). Alternatively, the mass spectra in the analog mode were recorded by scanning automatically the delay between the lasers using a LabView program, which was coded to operate synchronously the delay generators and the digital oscilloscope. Typically, around 30 delay scans were average in analog mode. Kinetic experiments using the analog output are illustrated in Supplementary Fig. 3 (Exps. #2 and #3).

**Resonance fluorescence.** Additional experiments to study the influence of water on the removal of atomic iodine were carried out with the Resonance and off-Resonance Fluorescence by Lamp Excitation (ROFLEX) machine[35] in an atmospheric pressure BBP flow tube set up. A 3–15 slm flow of synthetic air carrying molecular iodine (~$10^{10}$ molecule $cm^{-3}$) and ozone (~$10^{12}$ molecule $cm^{-3}$) was passed to a 4-cm-diameter, 50-cm-long quartz tube where it was irradiated by a Xe arc lamp. The main flow and flow ratios were varied to keep the concentration of $I_2$ and $O_3$ constant while varying the residence time of the gas mixture on the tube. The gas mixture was sampled at the end of the flow tube by using a small aperture into the ROFLEX chamber at 40 Torr. Iodine atomic resonance fluorescence is excited by a temperature-stabilized iodine radiofrequency lamp, and collected perpendicularly by using a VUV-sensitive channel photomultiplier in counting mode. The data collected in these experiments is plotted in Supplementary Fig. 4. The analysis of this dataset using numerical kinetic modeling is explained below.

**Kinetic analysis.** The processed PI-ToF-MS data consist of integrated peak signals (proportional to concentration) versus laser delay time in the case of PLP experiments, or a single mass spectrum in the case of BBP experiments, which correspond to a single time point, time being the residence time in the flow tube. The data obtained with the ROFLEX machine comprises the iodine atomic resonance fluorescence signal versus residence time in the flow tube.

Simple chemical systems can be described by sets of ordinary differential equations (ODEs) that are analytically solvable. In these cases, the observed data can be fitted to analytical expressions using a least-squares method, from which reaction rates can be extracted. This can be done for example in the case of the resonance fluorescence experiments if we assume that the only relevant reactions describing the behavior of atomic iodine are:

$$I_2 + h\nu \rightarrow I + I \tag{5}$$

$$I + O_3 \rightarrow IO + O_2 \tag{6}$$

$$I \rightarrow loss. \tag{7}$$

Because of the fast photolytic removal of $I_2$, an effective iodine loss rate in the presence of water can be extracted by simply fitting exponential decays to the time traces and calculating the differences between the removal rates with and without water. This, however, would not allow the concentration of $O_3$ to be estimated because IO reactions recycling atomic iodine are ignored:

$$IO + IO \rightarrow I + OIO \tag{8}$$

$$IO + h\nu \rightarrow I + O \tag{9}$$

$$O + O_2 + M \rightarrow O_3 + M. \tag{10}$$

Multiplexed detection systems like the PI-ToF-MS provide simultaneous data from many molecules. To explain their kinetic traces, complex mechanisms with many reactions need to be constructed (e.g., Supplementary Tables 1 and 2). For

these, the corresponding ODEs are coupled and not amenable to analytical solution. By combining an ODE numerical integrator and a nonlinear least-squares algorithm to fit simulated curves to the observed ones, some of the unknown reaction rates of the system may be determined, provided that the problem has enough degrees of freedom and that the free parameters are uncorrelated. When the initial conditions are known (concentration of precursors and laser energies), it is possible to derive the scaling factors of the signals observed by setting them as floating parameters. We use a standard integrator for stiff ODE problems (i.e., containing reaction rates spanning orders of magnitude) and a constrained nonlinear multivariable least-squares method from the Mathematics Matlab toolboxes[50].

The left panel of Supplementary Fig. 4 shows the results obtained for the ROFLEX flow tube data with the numerical integration–nonlinear fitting analysis considering IO reactions. The right panel compares the effective removal rate constants obtained by analytical fitting and numerical integration–nonlinear fitting.

**Iodine oxide particle pyrolysis**. The set up was modified to conduct a set of BBP-pyrolysis experiments by placing near the sampling point an iron wire shaped into a 10-mm diameter coil connected to a power supply by electrical feedthrough (Supplementary Fig. 2). The IOPs grew to a few nanometers[26] in the flow tube at room temperature over ~3 s residence time. The carrier gas and particles passed through the first wire loop and were then sampled from the center of the coil in order to detect potential evaporation products. The color of the glowing filament suggested a temperature range of 1500–2500 K, although the gas temperature was presumably much lower. The gas spent a few tens of milliseconds in the hot region before sampling. Before adding water to the main flow, the current through the wire was changed until all $I_xO_y$ signals were removed by thermal dissociation, in order to minimize potential contributions of gas-phase reactions generating $HOIO_2$ at high temperature.

**Electronic structure and master equation calculations**. All gas-phase quantum-mechanical calculations reported in this work were performed using the Gaussian09[51] software, and thermodynamic quantities are reported for the conditions of standard temperature (298.15 K) and pressure (1 atm).

The $I_2O_2 + H_2O \leftrightarrow HOIO + HOI$ and $I_2O_3 + H_2O \leftrightarrow HOIO_2 + HOI$ reactions in the gas-phase (Supplementary Figs. 5 and 6, respectively) were explored using a high-level method employed in previous work for the $I_2O_4 + H_2O \leftrightarrow HOIO_2 + HOIO$ (Supplementary Fig. 7) and $I_2O_5 + H_2O \leftrightarrow 2HOIO_2$ reactions[30]. The geometries of the stationary points on the PES of both reactions were first optimized at the M06-2X[52] level of theory, using the aug-cc-pVDZ[53] basis set for H and O, and the effective core potential LANL2DZ basis set for Iodine atoms (M06-2X/aug-cc-pVDZ+LANL2DZ). Normal-mode vibrational frequency analyses were carried out to ensure that the stable minima had all positive vibrational frequencies. Energies were refined at the CCSD(T)[54] level of theory with the aug-cc-pVTZ +LANL2DZ basis set. For all reactions, unscaled vibrational frequencies calculated with the M06-2X/aug-cc-pVDZ+LANL2DZ method were used to estimate the zero-point energy (ZPE) corrections.

A detailed description of the methods employed in our Born–Oppenheimer Molecular Dynamics (BOMD) simulations can be found in a previous publication[30]. The droplet system contained 191 water molecules and one $I_2O_y$ ($y$ = 2, 3, 4) molecule. The simulations were carried out in the constant volume and temperature (300 K) ensemble. The integration step was set as 1 femtosecond, which has been proven to achieve sufficient energy conservation in water systems[55–58]. The $I_2O_2$, $I_2O_3$, and $I_2O_4$ systems were simulated for 50, 80, and 33 picoseconds, respectively (Supplementary Videos 2, 3, and 4, respectively).

Estimates of vertical ionization and fragmentation energies of iodine oxides and oxyacids (Supplementary Table 2) as well as reaction enthalpies of relevant reactions (Table 1 and Supplementary Table 3) that are not available in the literature were obtained by using the B3LYP functional combined with the standard 6-311 + G(2d,p) triple-ζ basis set for O and H and an all-electron (AE) basis set for I[59]. This may be described as a supplemented (15s12p6d)/[10s9p4d] 6-311G basis, the [5211111111, 411111111, 3111] contraction scheme being supplemented by diffuse s and p functions, together with d and f polarization functions. The full basis set is given in Table XIII of Glukhovtsev et al.[59] Following geometry optimizations of the reactant and product molecules and ions and the determination of their corresponding vibrational frequencies and (harmonic) ZPE, energies relative to the reactants were obtained. Geometry optimizations and energy calculations for most of the molecules listed in Table 1 are already available from our previous work[32,60]. The additional calculations in this work have been carried out for $I_xO_y$ with $y$ = 3, 4 and $HIO_z$ with $z$ = 2, 3. Ionization energies and reaction enthalpies obtained at B3LYP/6-311 + G(2d, p) (+AE) level of theory compare reasonably well with experimental values and calculations at the higher level reported in the literature (see Table 1 and Supplementary Tables 2 and 3). This method has also been used to calculate the enthalpies of chemical ionization reactions of iodine oxides and oxyacids with nitrate, bromide and acetate ions generating $IO^-$, $IO_2^-$, and $IO_3^-$ (Supplementary Table 3).

To determine rate constants from ab initio data, we have used the Master Equation Solver for Multi-Energy well Reactions (MESMER)[39,61]. The set of ro-vibrational energy levels in the ab initio PES are grouped into energy grains, whose populations are described by a system of coupled differential equations that

account for collisional energy transfer and dissociation. The microcanonical rate coefficients of the unimolecular reactions that occur in each grain are calculated from the ab initio PES. For barrierless association reactions, the inverse Laplace transform method (ILT)[62,63] is used to calculate the microcanonical association rates from an estimate of the high-pressure limiting-rate coefficient for the molecular association. The microcanonical dissociation rate coefficients are then determined by detailed balance. Collisional energy transfer probabilities are described by using the exponential down model. For further details, see Galvez et al.[32] and references therein.

## Data availability
The data that support the findings of this study are available from the corresponding authors on reasonable request.

## Code availability
The LabView and Matlab software tools written to respectively synchronize the experiments and analyze the kinetic data are available from the corresponding authors on reasonable request.

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

## Acknowledgements

This study has received funding from the European Research Council Executive Agency under the European Union's Horizon 2020 Research and Innovation programme (Project "ERC-2016-COG 726349 CLIMAHAL"). J.C.G.M. acknowledges financial support from the State Agency for Research of the Spanish MCIU through the "Center of Excellence Severo Ochoa" award to the Instituto de Astrofísica de Andalucía (SEV-2017-0709), the Ramon y Cajal Program (RYC-2016-19570) and the National I + D + i Program (RTI2018-095330-B-100). We thank the Holland Computing Center of the University of Nebraska—Lincoln for providing computing resources.

## Author contributions

A.S.-L. devised the research. J.C.G.M., M.A.B., and J.M.C.P. designed the experimental set up, and J.C.G.M. and T.R.L. carried out the experiments and analyzed the data; J.M.C.P., J.C.G.M., M.K., and J.S.F. carried out electronic structure and master equation calculations; J.C.G.M. wrote the paper with contributions from all co-authors.

## Competing interests

The authors declare no competing interests.
