## [Peer Review File · Nature Communications]

Reviewer Comments, first round:

Reviewer #1 (Remarks to the Author):

Particle formation by gas-particle conversion is an important process that determines the concentration of atmospheric aerosols and ultimately the concentration of cloud condensation nuclei. Especially natural particle formation pathways are of particular interest, since they also form the basis for understanding the particle formation process influenced by humans. Studies of maritime processes, such as the formation of new particles from released iodine compounds, are of particular interest here, especially in order to understand the effects and feedback effects of global climate change. I therefore consider the present study to be an important contribution to atmospheric research that should be published in Nature Communications, Nature Research. The authors put together a novel and convincing data set, especially due to new and comprehensive data from laboratory studies, which is of interest for a broader scientific community in the atmospheric sciences. The major conclusion is that iodine oxyacids, which have been assumed to be the major gas phase precursors before, are unlikely to be major nucleating species of IOPs and that future modelling work should adapt the presented chemical scheme to carry out global assessments of the contribution of iodine new particle formation to the atmospheric aerosol burden and its associated radiative forcing. The manuscript is well written and generally clearly presented. Previous work is adequately cited. Therefore, I suggest to publish the manuscript after considering the following comments and suggestions.

The authors conclude that the attachment of water molecules to molecular clusters may have the effect of reducing cluster formation and particle coagulation rates and that the compositional change results in a change in cluster properties. This is a very general interpretation and I could imagine that a more fundamental discussion of the effect of cluster/nanoparticle composition, even if just based on theories, would help the general reader to understand the relevance of the submitted experimental study. Some very detailed information from the result section could, in my opinion, be included in the supplemental material.

Reviewer #2 (Remarks to the Author):

Gomez Martin et al present a comprehensive combined laboratory and theoretical modeling study of iodine cluster formation leading to new particle formation. The manuscript is well-written and provides compelling evidence as to the formation of iodine-containing particles, which is an important topic. Prior to this work, the initial steps of this process have remained uncertain, despite several measurements of iodine-containing clusters and ultrafine particles in marine environments globally. The manuscript would benefit from some clarifications, as described below, and especially additional discussion of the derivation of the mechanism shown in Figure 6.

The major outcome of the research presented is the mechanism shown in Figure 6, as it summarizes the combined findings of all of the results presented and integrates this with other literature. However, currently, there is little discussion of the mechanism in the main text (just referred to in the second to last sentence of the main text), which is a significant weakness of the paper. As a result, it is not clear how it was derived. The reader needs the authors to step through the mechanism to understand how it was compiled together (answering the question - what results contributed to each component?). This should be done in the Discussion section, but it would also be useful to refer to earlier in the text between discussing specific IxOy clusters, for example. It is key for this work that the reader understand completely where each step of the mechanism is derived from and where uncertainties exist. Additionally, since not all of the mechanism is derived from the current work (e.g. emission sources of iodine-containing trace gases), additional text and references should be provided in the figure caption. Also – has HOI actually been measured to be emitted from any of these sources? Additionally, does this mechanism explain the previously

measured iodine-containing clusters by Sipila et al?

Pages 2, 7, 8, & SI Section 7: The intro paragraph discusses previous measurements of iodine-containing clusters in the marine boundary layer and challenges, in the current literature, in the interpretation of these prior data. It is stated (Page 2, paragraph 1, Lines 7-8; Page 8, Paragraph 2, Lines 3 & 7) that IO₃⁻ and HIO₃⁻ containing cluster anions were observed by CI-API-ToF-MS. For a reader not familiar with the method, it is important to explain that these anions were formed from reaction with a reagent ion (i.e. formed during measurement). Also, given the discussion that HOIO₂ shouldn't form in the gas phase, the authors should acknowledge in this paragraph that non-clustered IO₃⁻ was measured (interpreted as HIO₃(g) prior to reaction using two reagent ions – nitrate and acetate). Similarly, on page 7, when the authors state that "HOI and HOIO₂...are not present in the gas phase and therefore are not indispensable particle precursors", it should be clarified that they are discussing their own results, which differ from the previous ambient measurements. Also, in Section 7 of the SI, the authors discuss possible interpretation of IO₃⁻ from nitrate CI, but they do not discuss IO₃⁻ from acetate CI. Addition of an examination of acetate reaction products would be useful here, especially since the authors are challenging interpretation of this previous work.

Page 5, Paragraph 1; Figure 4; & Methods: At what temperature did the HOIO₂ evaporate from the particles? What is the difference between "warm filament", "hot filament", "at low temperature", and "intense heat" (phrasing in Figure 4 legend and caption)? More information is needed in the methods section about the resistive heating method. Please also provide context in the Figure 4 caption as it currently seems contradictory (low temperature = warm filament??). Also, what does "when the temperature is sufficiently increased" mean? In the main text, the accompanying discussion needs to be augmented. Could HOIO₂ be formed during the heating process (addition of heat promoting a reaction), or could the HOIO₂ be a thermal decomposition product? Are you also suggesting that HOIO₂ and HOI could be in equilibrium with the gas-phase in the ambient atmosphere, explaining the previous measurement of HOIO₂? Greater description is needed in this section and its accompanying methods paragraph.

Additional Detailed Comments:

- Abstract: I suggest changing the wording "gas-phase HOIO₂ is not a precursor of HOIO₂-containing particles" to "gas-phase HOIO₂ is not necessary for the formation of HOIO₂-containing particles" to acknowledge the continued uncertainties in the mechanism, even following this work.

- Page 2, Paragraph 3: Add a statement of the benefits and caveats of using 248 nm radiation here since this is below the solar radiation spectrum.

- Page 3, Paragraph 1: Add a statement explaining/interpreting why I₂O₅⁺ is absent in this experiment.

- Page 4, Paragraph 1: The statement "...water partially inhibits the formation of large clusters and/or stabilizes them" is confusing, as it seems contradictory. Please clarify.

- Page 4, Paragraph 2: This paragraph discusses two experiment types, and to reduce confusion, it would benefit from being split into two, with the second paragraph starting with "We have also performed..." on line 7 of this paragraph.

- Page 5, Paragraph 3: This paragraph is completely discussion with no results. Move and incorporate this in the discussion section.

- Page 6, Paragraph 2, Lines 9-12: These sentences are confusing as presented, as they seem contradictory presenting a hypothesis but then seemingly ruling it out. The phrasing here can be clarified. Also, did the previous work that you are discussing use a long inlet?

- Page 7, Paragraph 3, Lines 8-10: This statement about precursor emission in the polar MBL is not from this work, and therefore, it needs a reference.

- Page 8, Line 2; Figure 2 caption, Line 4; Figure 3 caption, Line 4: Fix typos.

- Page 10, Paragraph 1: What size IOPs were grown? How long were they collected on the filament. Provide information about the temperature. Overall, more method information is needed, as these were key experiments, and yet few details about methodology are provided.
- Figure 1a: The different offsets of the blue and red traces are confusing as presented.
- Figure 3: Please improve the contrast/presentation of this plot to make it easier to view the signals. Not only are the signals difficult to view, but the black text on the dark blue background is also difficult to view. The authors might also consider moving this figure to the SI and bringing Figure S2 to the main text.
- Figure 4: Please explain in the caption what the single and double lines under the mass spectra correspond to.
- Figure 5: Could there be differences observed because the higher pressure experiment was completed with a PI of 10.5 eV, compared to the 11.6 eV for the lower pressure experiment, since HOIO₂ is only observed at 11.6 eV. This consequence needs to be noted/discussed.

Reviewer #3 (Remarks to the Author):

Review of "Connecting the dots of atmospheric iodine gas-to-particle conversion" by Gomez-Martin et al.

The study is exceptionally well-motivated. Nucleation and subsequent formation of viable new particles from coastal and marine iodine emissions has been the subject of intense research since the 1990s and recent work has rather contentiously interpreted the field observation of iodate and HIO₃ containing clusters as the sequential addition of gaseous HIO₃. The identification of molecules in the gas phase that lead to the initial clustering and the mechanism for their formation is the critical step in being able to predict the formation rate of new particles. Evidence related to the known (OH oxidation of iodine dioxide) and postulated (reaction of water vapour with iodine and its oxides) mechanisms for HIO₃ formation, would seem to preclude its generation at levels needed for particle formation. This study aims to systematically identify the products of iodine oxidation and new particle formation and test the hypothesis that HIO₃ is really the gas phase molecule responsible for the HIO₃ identified in the newly formed clusters.

The study has been carefully constructed and meticulously conducted. The combination of tested and proven techniques, building on known and well-established gaseous kinetics is perfectly well suited to answering the questions posed by the previous interpretation. The primary result that "water reacts very slowly with atomic iodine and iodine oxides, and that forming particulate HOIO₂ does not require gas-phase HOIO₂" and the implication that there is a "limited role of oxyacids in IOP formation, which is instead initiated by clustering of IxOy" appears to be based on a solid foundation and leaves little room for ambiguity. It is noteworthy that HIO₃ is not observed in dry experiments, nor in those with added water. Furthermore, the absence of I₂O₅ is perhaps surprising, but clear from the experiments and similarly noteworthy.

The interpretation in the Sipilä et al (2016) Nature study should be viewed in context of the statement in the current work that "a succession of peaks in a mass spectrum does not necessarily reveal how a nucleation mechanism works". Results from the time resolved experiments required to resolve the nucleation mechanism are convincing and clear. Most importantly "...the time resolved mass spectra obtained for humid conditions are almost identical to those obtained under dry conditions, except for the presence of HOI and a slight decrease of all IxOy signals". It is important that neither HIO₃ nor HIO₃-containing peaks are observed in either pulsed laser or broadband photolysis experiments, which would resolve products of slower reactions. It appears water does not help form oxyacids (and indeed seems to quench particle formation in the broadband experiments). The current work then proceeds to identify oxyacid mass spectral peaks in experiments subjecting particles to resistive heating. These appear to be the evaporation

products in the presence of water, providing a plausible mechanism for their formation. The last piece of the jigsaw is provided by the ab initio calculations coupled with the kinetic modelling that demonstrate the thermochemical difficulty in producing oxyacids under reasonable ambient conditions. This is important, since it indicates that reaction on walls in e.g. long residence-time experiments and long field inlet lines, may produce oxyacids.

The mechanism for photo-oxidation and subsequent clustering necessary to explain the experiments is insightful and useful, if only tentative. This is to be expected, since the system is still underconstrained (particularly owing to cluster fragmentation). The exceptionally high rate constants (beyond the kinetic limit) for higher oxide formation clearly require long-range attraction owing to the very high molecular dipole moments. This system is clearly complex and in future work might test whether a model including all possible collision partners would optimise to the same mechanism given the experimental constraint in the current paper (probably using machine learning approaches), or whether there are multiple plausible solutions. In any case, the current mechanism is the first to convincingly and explicitly connect higher iodine oxides and new particle clusters.

I have few criticisms of the work and it will be a valuable contribution to the literature. I fully recommend it for publication.

Just a couple of comments: first, it is stated in the abstract that HIO₃ is "the currently accepted nucleating molecule". I'd suggest that it has been relatively recently-postulated and the identification is rather tentative and by one instrumental technique (CI-API-ToF-MS) in Sipilä et al. (2016). It was the subject of a high profile publication, but I'd not agree that it is currently accepted.

Probably more importantly, I'd suggest a less colloquial and more self-explanatory title - it is not clear what the "dots" are. In addition to the piecing together of the puzzle, I presume it relates to the "dots" in the mass defect plot 1b) and that in the previous Sipilä et al. paper, but it's probably better suited to a subtitle than a main title.

Paper Ref: NCOMMS-20-10505-T

Title: "Connecting the dots of atmospheric iodine gas-to-particle conversion" -> "**New insights into atmospheric iodine gas-to-particle conversion**"

RESPONSE TO THE REVIEWERS' REPORTS

We are grateful to the reviewers for helpful and constructive comments and suggestions. We address them point by point below. Reviewers' comments are shown in **bold typescript**, our response in normal typescript. Additions to the manuscript are highlighted in red. Page numbers refer to the revised manuscript.

Reviewer #1 (Remarks to the Author):

Particle formation by gas-particle conversion is an important process that determines the concentration of atmospheric aerosols and ultimately the concentration of cloud condensation nuclei. Especially natural particle formation pathways are of particular interest, since they also form the basis for understanding the particle formation process influenced by humans. Studies of maritime processes, such as the formation of new particles from released iodine compounds, are of particular interest here, especially in order to understand the effects and feedback effects of global climate change. I therefore consider the present study to be an important contribution to atmospheric research that should be published in Nature Communications, Nature Research. The authors put together a novel and convincing data set, especially due to new and comprehensive data from laboratory studies, which is of interest for a broader scientific community in the atmospheric sciences. The major conclusion is that iodine oxyacids, which have been assumed to be the major gas phase precursors before, are unlikely to be major nucleating species of IOPs and that future modelling work should adapt the presented chemical scheme to carry out global assessments of the contribution of iodine new particle formation to the atmospheric aerosol burden and its associated radiative forcing. The manuscript is well written and generally clearly presented. Previous work is adequately cited. Therefore, I suggest to publish the manuscript after considering the following comments and suggestions.

The authors conclude that the attachment of water molecules to molecular clusters may

have the effect of reducing cluster formation and particle coagulation rates and that the compositional change results in a change in cluster properties. This is a very general interpretation and I could imagine that a more fundamental discussion of the effect of cluster/nanoparticle composition, even if just based on theories, would help the general reader to understand the relevance of the submitted experimental study.

Evidence for reduced cluster formation upon addition of water stems from the reduction of the mass spectrometric signals of large molecular clusters, as well as from the direct observation through the viewports of the flow tube of a drastic reduction of light scattering by iodine smoke particles produced inside the reactor in the continuous broad band photolysis experiments. As discussed on p. 7, this is in line with our previous observations of particle formation inhibition by water in an aerosol flow tube experiment using a nano-differential mobility particle sizer (Saunders et al., 2010), and it is also consistent with field observations during the PARFORCE campaign showing that the probability of nucleation events decreases substantially with increasing humidity (de Leeuw et al. 2002). As stated on p. 7, the attachment of water molecules to molecular clusters may have the effect of reducing cluster formation and particle coagulation rates. Saunders et al. 2010 and Galvez et al. 2013 argued that the key iodine oxide I_2O_4 forms a relatively stable complex with water, such that under the conditions prevailing in the marine boundary layer, a large fraction of I_2O_4 molecules would be hydrated as $I_2O_4 \cdot H_2O$ (>40%). This could block the I_2O_4 halogen bonding sites and the subsequent necessary geometrical rearrangement for the reaction of I_2O_4 with itself or with I_2O_3 to proceed may result in barriers in the corresponding potential energy surface (there are some examples of this, such as $BrO + HO_2$ and $HNO_3 + OH$). The same may occur for larger molecular clusters. It must be noted that $I_xO_y \cdot H_2O$ clusters are not observed in our experiments, which may be because they cannot be detected by VUV-PI, e.g. because they may fall apart upon absorption of VUV photons. Thus, this explanation remains to be tested in future studies.

Changes:

P. 7, insertion: “We found previously^{26,37} that the key iodine oxide I_2O_4 forms a relatively stable complex with water, such that under the conditions prevailing in the MBL, a large fraction of I_2O_4 molecules would be hydrated as $I_2O_4 \cdot H_2O$ (>40%). This could block the I_2O_4

halogen bonding sites and the subsequent necessary geometrical rearrangement for the reaction of I_2O_4 with itself or with I_2O_4 to proceed may result in barriers in the corresponding potential energy surface. The same may occur for larger molecular clusters. It should be noted that $I_xO_y \cdot H_2O$ clusters are not observed in our experiments, which may be because they promptly dissociate following absorption of VUV photons.”

Some very detailed information from the result section could, in my opinion, be included in the supplemental material.

Since this is essentially an experimental study, we feel it is necessary to describe the experimental set up with some level of detail, so that reading the supplementary material is not strictly necessary for the general reader. Moreover, the methods section contains 828 words, which is well within the range permitted by Nat Comms (<3000).

Reviewer #2 (Remarks to the Author):

Gomez Martin et al present a comprehensive combined laboratory and theoretical modeling study of iodine cluster formation leading to new particle formation. The manuscript is well-written and provides compelling evidence as to the formation of iodine-containing particles, which is an important topic. Prior to this work, the initial steps of this process have remained uncertain, despite several measurements of iodine-containing clusters and ultrafine particles in marine environments globally. The manuscript would benefit from some clarifications, as described below, and especially additional discussion of the derivation of the mechanism shown in Figure 6.

The major outcome of the research presented is the mechanism shown in Figure 6, as it summarizes the combined findings of all of the results presented and integrates this with other literature. However, currently, there is little discussion of the mechanism in the main text (just referred to in the second to last sentence of the main text), which is a significant weakness of the paper. As a result, it is not clear how it was derived. The reader needs the authors to step through the mechanism to understand how it was compiled together (answering the question - what results contributed to each component?). This should be done in the Discussion section, but it would also be useful to refer to earlier in the text between discussing specific I_xO_y clusters, for example. It is key

for this work that the reader understand completely where each step of the mechanism is derived from and where uncertainties exist.

The derivation of the mechanism is briefly summarized in the 7th paragraph of the results section of the main text (p. 5), and fully described in section 6 of the supplementary information. The link between the mechanism deduced from laboratory data and the basic IOP formation mechanism proposed in Figure 7 (previously Figure 6) is mentioned in the first sentence of the discussion section. However, a reference to Figure 7 was missing, which may have led the reviewer to think that the mechanism is only mentioned in the last sentences of the main text.

A very detailed discussion of how the mechanism is derived cannot be included in the main text due to manuscript length constraints (<5000 words). However, we believe that the following additional sentences should help the reader to follow how the sequence proposed in Figure 7 is firmly grounded in the mechanism required to explain our set of laboratory results.

Changes:

Results section, p. 5, insertions: “The rate constants of the selected reactions are then adjusted to model the fast formation and decay of the I_xO_y species **as shown in Figure 4 (the method is described in the Supplementary Material).**”

Results section, p. 6, insertions: “In this I_xO_y nucleation mechanism, rapid pressure-independent bimolecular reactions between I_xO_y molecular clusters form larger clusters with **the observed** odd numbers of iodine atoms ($(I_2O_5)_nOIO$), e.g.:

Additionally, third body reactions at their high pressure limit form aggregates, e.g.:

where the O/I ratio of the aggregates may be subsequently increased through oxidation by O_3 . The specific reactions considered in the example shown in Figure 4 and their estimated

rate constants are listed in Table S5.”

Discussion section, p. 7, insertions: “Our laboratory results strongly support an atmospheric gas-to-particle conversion mechanism where the initial clustering steps are driven by I_xO_y bimolecular and third body reactions, both under dry and wet conditions. The mechanism of atmospheric IOP formation proposed in Figure 7 is based on the set of reactions (Table S5) that best reproduces the time-resolved mass spectrometric data (Figure 4) in the presence and absence of water.”

Discussion section, p 7-8, insertion: “This evidences the transformation of the I_xO_y clusters into oxyacid-containing clusters within the flow tube residence time, which confirms earlier predictions about the composition of I_2O_5 IOPs in the MBL²⁸. This scenario is represented in Figure 7 by the hydration of the I_2O_5 units present in the iodine oxide clusters with composition $(I_2O_5)_nOIO$.”

Additionally, since not all of the mechanism is derived from the current work (e.g. emission sources of iodine-containing trace gases), additional text and references should be provided in the figure caption.

Changes:

Caption Figure 6 (now Figure 7): “**Figure 7.** Proposed mechanism of iodine oxide new particle formation. Iodine-bearing molecules are emitted from the ocean surface and marine biota⁴⁵⁻⁴⁸ and photo-oxidized in the presence of O_3 , leading to the formation of OIO ³³ and I_2O_y molecules²⁷. According to the findings of the present work, I_2O_y reactions lead to the formation of molecular clusters with composition $OIO(I_2O_5)_n$, which transition into $HOIO_2$ -containing clusters by progressive hydration. Further uptake of water, organic acids and H_2SO_4 ²⁶ leads to particle formation. The formation of iodic acid ($HOIO_2$) by reaction of H_2O with I_2O_y cannot be disproved, but appears to be too slow for $HOIO_2$ nucleation to occur. It is likely that $HOIO_2$ formed in this way - or from the $OIO + OH$ - reaction is taken up by IOP clusters and aerosol. ”

Also – has HOI actually been measured to be emitted from any of these sources?

Yes, I_2 and HOI are the gas phase products of the heterogeneous reaction between iodide ions in water-air interfaces (sea and aerosol surfaces) and $O_3(g)$ (Carpenter et al. 2013,

MacDonald et al. 2014)

Additionally, does this mechanism explain the previously measured iodine-containing clusters by Sipilä et al?

The negative ion clusters observed by Sipilä et al. can be viewed as consisting of two progressions with compositions $(I_2O_5)_nIO_3^-$ and $(I_2O_5)_n(HIO_3)IO_3^-$. Clusters with a small number of missing oxygen atoms with respect to these compositions also appear. Sipilä et al explained these two progressions as the result of sequential $HOIO_2$ addition and water elimination steps. In our experiments, we observe clusters with the general composition $(I_2O_5)_nOIO^+$ (i.e. no clusters with even number of iodine atoms), and also some clusters with missing iodine atoms. Our mechanism is built to explain these clusters from the reaction of iodine oxide precursors and does not attempt to explain the molecular clusters observed by Sipilä et al., which we do not observe. We have emphasized the disagreement and the exclusion of these clusters from the proposed mechanism.

Changes:

P. 3, insertion: “(e.g. $(I_2O_5)_n(HIO_3)_{0-2}$ ²³)”

P. 5, insertion: “ $HOIO_2$ -containing clusters ²³ have not been observed and therefore no attempt has been made to model them.”

Pages 2, 7, 8, & SI Section 7: The intro paragraph discusses previous measurements of iodine-containing clusters in the marine boundary layer and challenges, in the current literature, in the interpretation of these prior data. It is stated (Page 2, paragraph 1, Lines 7-8; Page 8, Paragraph 2, Lines 3 & 7) that IO_3^- and HIO_3^- containing cluster anions were observed by CI-API-ToF-MS. For a reader not familiar with the method, it is important to explain that these anions were formed from reaction with a reagent ion (i.e. formed during measurement). Also, given the discussion that $HOIO_2$ shouldn't form in the gas phase, the authors should acknowledge in this paragraph that non-clustered IO_3^- was measured (interpreted as $HIO_3(g)$ prior to reaction using two reagent ions – nitrate and acetate).

Changes:

P. 2, insertion: “These observations, which employ nitrate (NO_3^-) and acetate (CH_3COO^-) chemi-ionization sources, have been interpreted as resulting from atmospheric gas phase HOIO_2 and molecular cluster formation via HOIO_2 addition steps.”

Discussion section, p. 8, insertion: “we have evidence indicating that Cl-API-ToF-MS IO_3^- signals attributed to gas-phase HOIO_2 prior to reaction using two reagent ions – nitrate and acetate – may require some re-interpretation”

Discussion section, p. 9, rewording: “Besides collisional fragmentation issues of this technique³⁹, IO_3^- ions are exothermic reaction products of the dissociative charge transfer between $\text{I}_2\text{O}_\gamma$ ($\gamma = 2, 3, 4$) and NO_3^- (and CH_3COO^-), and therefore it is likely that $\text{I}_2\text{O}_\gamma$ contributes significantly to the IO_3^- signal.”

Supplementary Information, section 7, p. 13, insertion: “The IO^- , IO_2^- and IO_3^- signals with and without added water have been attributed to the chemical ionization by nitrate and acetate ions of the corresponding oxyacid neutral molecules⁵”

Similarly, on page 7, when the authors state that “HOI and HOIO_2 ...are not present in the gas phase and therefore are not indispensable particle precursors”, it should be clarified that they are discussing their own results, which differ from the previous ambient measurements.

Changes:

Discussion section, p. 7, inserted: “The IOP pyrolysis experiments show that HOI and HOIO_2 are indeed constituents of these particles, even though we did not observe them in the gas phase at room temperature, which implies that they are not indispensable IOP gas-phase precursors.”

Also, in Section 7 of the SI, the authors discuss possible interpretation of IO_3^- from nitrate CI, but they do not discuss IO_3^- from acetate CI. Addition of an examination of acetate reaction products would be useful here, especially since the authors are challenging interpretation of this previous work.

Table S4 includes enthalpies of NO_3^- , Br^- and CH_3COO^- reactions with I_xO_y and HIO_x molecules resulting in IO_x^- products. The situation regarding IO_3^- is similar for these three types of CI

ion source.

Changes: We now refer to NO_3^- , Br^- and CH_3COO^- ionization in the discussion of possible reinterpretation of Cl IO_3^- signals section 7 of the SI. We also refer to nitrate and acetate in the main text, p. 8 (see above).

Page 5, Paragraph 1; Figure 4; & Methods: At what temperature did the HOIO₂ evaporate from the particles? What is the difference between “warm filament”, “hot filament”, “at low temperature”, and “intense heat” (phrasing in Figure 4 legend and caption)? More information is needed in the methods section about the resistive heating method. Please also provide context in the Figure 4 caption as it currently seems contradictory (low temperature = warm filament???). Also, what does “when the temperature is sufficiently increased” mean?

It was not possible to measure the temperature at the sampling point due to geometrical constraints. The colour of the glowing filament suggests a temperature range of 1500 K-2500 K, but the temperature at the sampling point is essentially unknown. We have changed the legend of Figure 4, which now refers to filament current rather than temperature.

The current through the heating coil was ramped up until a HOIO₂ signal was observed. No signal was observed at 1.7 A, while at 1.9 A some signal started to be clear over the background noise. In the particular case of the examples shown in Figure 4, the temperature was such that, without added water, iodine oxide masses larger than OIO were no longer observed, while there was concurrently an increase of the I, IO and I₂ signals. We observed the IOP smoke surrounding the hot filament to break down, which indicates that the IOPs were actually vaporised.

These observations can be explained by:

- 1) Dissociation of I_xO_y molecules with x>2, e.g. I₂O₄ (Saunders et al., 2010).
- 2) Thermal decomposition of solid I₂O₅ to gas-phase I₂ and O₂ occurs (T > 500 K). Other solid iodine oxides like I₂O₄ decompose to I₂O₅ at lower temperatures (Daehlie and Kjekshus, 1964; Selte and Kjekshus, 1968; Wikjord et al., 1980).

The removal of I_xO_y peaks at high temperature is explained both by molecular dissociation and reduction of fragmentation signal from larger clusters.

The mass spectra are similar when water is added at the same filament current, except for the emergence of oxyacid peaks. This suggests that in the presence of water, HOIO₂ forms on IOPs, and a fraction of it evaporates when the particles are heated.

Changes:

Results section, p. 5, several edits: “BBP of I₂/O₃ mixtures at room temperature generates I_xO_y and IOPs that travel for a few seconds down the flow tube towards the detection region. A few millimeters upstream of the sampling volume the carrier gas and the particles are resistively heated in order to observe evaporation products. The current through the filament was set to a value such that the I_xO_y molecules carried by the gas thermally dissociated in the absence of water (disappearance of the higher masses). Thus, the observed HOIO₂ and HOI cannot be products of gas-phase high temperature I_xO_y + H₂O reactions, but evaporation products of IOPs formed upstream in the presence of water. Under dry conditions, the oxyacid evaporation signals reduce drastically. The other effect of heating IOPs is the expected thermal decomposition of solid/liquid phase iodine oxides into molecular oxygen and iodine^{36,37} (enhancement of the I₂ signal).”

Caption Figure 4 (now Figure 5), insertions: “**Figure 4.** IOP pyrolysis experiments (PI energy 11.6 eV). The IOPs are grown along the flow tube and the carrier gas and particles are passed through a 1 cm diameter resistively heated filament coil just before sampling to detect evaporation products. At a lower filament current of 1.7 A (lower temperature at the sampling point), the spectrum obtained at 30 Torr with 2% water mixing ratio resembles other BBP spectra in Figure 2. When the current through the coil is higher (3.8 A), the spectrum changes (red line) as a result of IOP evaporation: the higher masses disappear, the I₂ signal increases and new masses show up at m/z = 144 (HOI⁺) and m/z = 144 (HOIO₂⁺). [...]”

In the main text, the accompanying discussion needs to be augmented. Could HOIO₂ be formed during the heating process (addition of heat promoting a reaction), or could the HOIO₂ be a thermal decomposition product?

When water was not added to the flow and the current through the filament was ramped up, the I₂O_y signals disappeared and there was an increase of the I and IO signals. This is most likely due to dissociation of I_xO_y molecules, which do not have very high binding

energies (Saunders et al. 2010, Kaltsoyannis and Plane, 2008, Galvez et al, 2013). Thus, the addition of water is not responsible for the removal of the I_xO_y signals. By contrast, IOx signals are not removed, and thus a possible source of HOIO₂ would be OIO + H₂O. However, this reaction is extremely endothermic (242 kJ mol⁻¹) and would be unlikely to produce a signal even at 2500 K. By contrast, the possibility that some HOI originates from IO + H₂O at high temperature cannot be entirely excluded.

The HOIO₂ signal observed at high temperature in the presence of water is then most likely generated from its evaporation from water processed IOP clusters.

Changes: see below

Are you also suggesting that HOIO₂ and HOI could be in equilibrium with the gas-phase in the ambient atmosphere, explaining the previous measurement of HOIO₂?

We are not suggesting that the liquid-gas HOIO₂ equilibrium is responsible for the CI atmospheric observations. We suggest in the discussion section that previous CI measurements can be interpreted differently as resulting from ionization of I₂O_y species by NO₃⁻ or CH₃COO⁻.

Greater description is needed in this section and its accompanying methods paragraph.

The discussion of the pyrolysis results and the description of the experimental set up have been expanded as requested, including a new Figure S2 in the Supplementary Material.

Changes:

Methods section, p. 10, several edits to the methods paragraph explaining the pyrolysis experiments:” The set up was modified to conduct a set of BBP-pyrolysis experiments by placing near the sampling point an iron wire **shaped into a 10 mm diameter coil** connected to a power supply by an electrical feedthrough (**Figure S2**). **The IOPs grew to a few nanometers²⁶ in the flow tube at room temperature over ~ 3 seconds residence time. The carrier gas and particles passed through the first wire loop and were then sampled from the center of the coil in order to detect potential evaporation products. The color of the glowing filament suggested a temperature range of 1500 K - 2500 K, although the gas temperature was presumably much lower. The gas spent a few tens of milliseconds in the hot region**

before sampling. Before adding water to the main flow, the current through the wire was changed until all I_xO_y signals were removed by thermal dissociation, in order to minimize potential contributions of gas phase reactions generating $HOIO_2$ at high temperature.”

New Figure S2 in the Supplementary Material:

Figure S1. Left: view of the resistively heated coil placed around the sampling point in the pyrolysis experiments. The picture was taken from a viewport perpendicular to the flow tube, situated where the power supply of the coil is depicted in Figure S1. Right: same picture with annotations indicating different elements of the set up and the sampling geometry.

Additional Detailed Comments:

- **Abstract:** I suggest changing the wording “gas-phase $HOIO_2$ is not a precursor of $HOIO_2$ -containing particles” to “gas-phase $HOIO_2$ is not necessary for the formation of $HOIO_2$ -containing particles” to acknowledge the continued uncertainties in the mechanism, even following this work.

Done.

- **Page 2, Paragraph 3:** Add a statement of the benefits and caveats of using 248 nm radiation here since this is below the solar radiation spectrum.

As mentioned in the methods section, several photochemical schemes are used in different experiments to generate iodine oxides. This may be by photolyzing O_3 at 248 nm in the presence of I_2 ($O + I_2 \rightarrow IO + I$) or by photolyzing I_2 at 193 nm or at ~ 500 nm in the presence of O_3 ($I + O_3 \rightarrow IO + O_2$). The specific pulsed laser source employed is not important for the conclusions of this paper as long as iodine monoxide is generated as precursor of I_xO_y in

detectable amounts.

- Page 3, Paragraph 1: Add a statement explaining/intepreting why I₂O₅+ is absent in this experiment.

As stated in p.3 , I₂O₅ is not observed at 11.6 eV, i.e. with PI photons above its ionization energy, and therefore we conclude that it does not form in the gas phase.

Changes:

Results section, p. 3, insertion: “Starting from I₃O₇⁺ and moving to higher m/z, the peaks are separated by I₂O₅ units (**Figure 1b**), **even though gas-phase I₂O₅ does not appear to form.**”

- Page 4, Paragraph 1: The statement “...water partially inhibits the formation of large clusters and/or stabilizes them” is confusing, as it seems contradictory. Please clarify.

There are two possible explanations of the reduction of I_xO_y peaks upon addition of water:

- 1) water complexes are formed, which create barriers in the PES of clustering reactions
- 2) water changes the composition of large clusters and makes them less prone to fragment by enhancing binding energy.

This second possibility is perhaps too speculative and not entirely compatible with the fact that iodine smoke is suppressed by addition of water, and therefore we have removed it from the text.

Changes:

Results section, p. 4, removed: “and/or stabilizes them”

Discussion section, p. 8, removed: “The compositional change evidenced by the appearance of oxyacids in clusters exposed to water seems to result in a change in their properties, e.g. an enhanced binding energy”

- Page 4, Paragraph 2: This paragraph discusses two experiment types, and to reduce confusion, it would benefit from being split into two, with the second paragraph starting with “We have also performed...” on line 7 of this paragraph.

Done.

- Page 5, Paragraph 3: This paragraph is completely discussion with no results. Move and incorporate this in the discussion section.

This paragraph was intended as an introduction to the ab initio results on $I_2O_y + H_2O$ reactions. Changes: the paragraph has been partially moved to page 2 where the known H_2O and HO_x chemistry is briefly introduced.

- Page 6, Paragraph 2, Lines 9-12: These sentences are confusing as presented, as they seem contradictory presenting a hypothesis but then seemingly ruling it out. The phrasing here can be clarified.

What we are doing here is comparing the upper limit rate constant set by the PI-ToF-MS results to a hypothetical $I+H_2O+O_3$ composite oxyacid source mechanism, to the rate constant obtained for the removal of atomic iodine in the presence of water and ozone obtained from our resonance fluorescence (ROFLEX) experiments. The comparison strongly suggest that this mechanism can be ruled out.

Changes:

P. 6, insertion: “The results obtained for a hypothetical composite reaction where atomic iodine complexes with water, and the resulting $H_2O...I$ adduct reacts with O_3 to form oxyacids (panels 6a and 6e) can be compared with the results obtained from the resonance fluorescence experiments on the removal of atomic iodine by O_3 in the presence of water vapor.”

Also, did the previous work that you are discussing using a long inlet?

We do not follow the reviewer here. In page 6, paragraph 2, we discuss the results from two experiments (PI-ToF-MS and ROFLEX) performed in the course of this work. The ROFLEX setup is briefly discussed in the Methods section and described in more detail in the supplementary material. It is a flow tube experiment with a similar configuration, if that is what the reviewer is asking.

- Page 7, Paragraph 3, Lines 8-10: This statement about precursor emission in the polar MBL is not from this work, and therefore, it needs a reference.

Done.

- Page 8, Line 2; Figure 2 caption, Line 4; Figure 3 caption, Line 4: Fix typos.

Done.

- Page 10, Paragraph 1: What size IOPs were grown? How long were they collected on the filament.

We did not measure the size distributions, but we were able to observe light scattering by iodine oxide smoke in our experiments, which confirms that particles were being made. Based on previous experiments in the same lab using a flow tube of similar length with a nano-DMA (Saunders et al 2010), we estimate the diameter of these particles to be up to a few nanometres.

The particles were not collected on the filament. The pyrolysis experiments were carried out online (no deposition time).

Changes: a more detailed discussion of the pyrolysis set up has been added to the methods section, including the sampling method (see above).

Provide information about the temperature. Overall, more method information is needed, as these were key experiments, and yet few details about methodology are provided.

See response above.

- Figure 1a: The different offsets of the blue and red traces are confusing as presented.

Changes:

Caption Figure 1, “shifted” changed to “with an offset”

Legend Figure 1, insertion : “+ offset”

- Figure 3: Please improve the contrast/presentation of this plot to make it easier to view the signals. Not only are the signals difficult to view, but the black text on the dark blue background is also difficult to view. The authors might also consider moving this figure to the SI and bringing Figure S2 to the main text.

The contrast in Figure 3 has been enhanced. The figure is kept in the main text since it provides an overall view of a full experiment. Following the recommendation of the reviewer, we have transferred Figure S2 to the main text (new Figure 4).

Other changes related to this point:

Results section, p. 3, insertion: “Laboratory time-resolved multiplexed experiments (Figure 3) [...]”

- **Figure 4: Please explain in the caption what the single and double lines under the mass spectra correspond to.**

Caption Figure 4 (now Figure 5), insertion: “[...] The black lines indicate the position of the oxide and oxyacids in the m/z axis: from left to right: I, IO, HOI, OIO, HOIO, HOIO₂, HIO₄, I₂, I₂O, I₂O₂, I₂O₃, I₂O₄, I₂O₅, HI₂O₅, I₃O₅, I₃O₆, I₃O₇.”

- **Figure 5: Could there be differences observed because the higher pressure experiment was completed with a PI of 10.5 eV, compared to the 11.6 eV for the lower pressure experiment, since HOIO₂ is only observed at 11.6 eV. This consequence needs to be noted/discussed.**

HOIO₂ is unlikely to be observed at 10.5 eV because this energy is below the calculated ionisation potential, as explained on page 3. Furthermore, HOIO₂ was not observed either in 10.5 eV experiments at lower pressure as shown in Figure 2. Some reactions are pressure dependent (e.g. the branching of the IO self-reaction), but this is considered in the modelling performed to find the upper limits to the rate constants.

Reviewer #3 (Remarks to the Author):

Review of "Connecting the dots of atmospheric iodine gas-to-particle conversion" by Gomez-Martin et al.

The study is exceptionally well-motivated. Nucleation and subsequent formation of viable new particles from coastal and marine iodine emissions has been the subject of intense research since the 1990s and recent work has rather contentiously interpreted the field observation of iodate and HIO₃ containing clusters as the sequential addition of gaseous HIO₃. The identification of molecules in the gas phase that lead to the initial clustering

and the mechanism for their formation is the critical step in being able to predict the formation rate of new particles. Evidence related to the known (OH oxidation of iodine dioxide) and postulated (reaction of water vapour with iodine and its oxides) mechanisms for HIO₃ formation, would seem to preclude its generation at levels needed for particle formation. This study aims to systematically identify the products of iodine oxidation and new particle formation and test the hypothesis that HIO₃ is really the gas phase molecule responsible for the HIO₃ identified in the newly formed clusters.

The study has been carefully constructed and meticulously conducted. The combination of tested and proven techniques, building on known and well-established gaseous kinetics is perfectly well suited to answering the questions posed by the previous interpretation. The primary result that "water reacts very slowly with atomic iodine and iodine oxides, and that forming particulate HOIO₂ does not require gas-phase HOIO₂" and the implication that there is a "limited role of oxyacids in IOP formation, which is instead initiated by clustering of IxOy" appears to be based on a solid foundation and leaves little room for ambiguity. It is noteworthy that HIO₃ is not observed in dry experiments, nor in those with added water. Furthermore, the absence of I₂O₅ is perhaps surprising, but clear from the experiments and similarly noteworthy.

The interpretation in the Sipilä et al (2016) Nature study should be viewed in context of the statement in the current work that "a succession of peaks in a mass spectrum does not necessarily reveal how a nucleation mechanism works". Results from the time resolved experiments required to resolve the nucleation mechanism are convincing and clear. Most importantly "...the time resolved mass spectra obtained for humid conditions are almost identical to those obtained under dry conditions, except for the presence of HOI and a slight decrease of all IxOy signals". It is important that neither HIO₃ nor HIO₃-containing peaks are observed in either pulsed laser or broadband photolysis experiments, which would resolve products of slower reactions. It appears water does not help form oxyacids (and indeed seems to quench particle formation in the broadband experiments). The current work then proceeds to identify oxyacid mass spectral peaks in experiments subjecting particles to resistive heating. These appear to be the evaporation products in the presence of water, providing a plausible mechanism for their formation. The last piece of the jigsaw is provided by the ab initio calculations coupled with the

kinetic modelling that demonstrate the thermochemical difficulty in producing oxyacids under reasonable ambient conditions. This is important, since it indicates that reaction on walls in e.g. long residence-time experiments and long field inlet lines, may produce oxyacids.

The mechanism for photo-oxidation and subsequent clustering necessary to explain the experiments is insightful and useful, if only tentative. This is to be expected, since the system is still underconstrained (particularly owing to cluster fragmentation). The exceptionally high rate constants (beyond the kinetic limit) for higher oxide formation clearly require long-range attraction owing to the very high molecular dipole moments. This system is clearly complex and in future work might test whether a model including all possible collision partners would optimise to the same mechanism given the experimental constraint in the current paper (probably using machine learning approaches), or whether there are multiple plausible solutions. In any case, the current mechanism is the first to convincingly and explicitly connect higher iodine oxides and new particle clusters.

I have few criticisms of the work and it will be a valuable contribution to the literature. I fully recommend it for publication.

Just a couple of comments: first, it is stated in the abstract that HIO₃ is "the currently accepted nucleating molecule". I'd suggest that it has been relatively recently-postulated and the identification is rather tentative and by one instrumental technique (CI-API-ToF-MS) in Sipilä et al. (2016). It was the subject of a high profile publication, but I'd not agree that it is currently accepted.

Abstract: "currently accepted" changed to "recently proposed".

Probably more importantly, I'd suggest a less colloquial and more self-explanatory title - it is not clear what the "dots" are. In addition to the piecing together of the puzzle, I presume it relates to the "dots" in the mass defect plot 1b) and that in the previous Sipilä et al. paper, but it's probably better suited to a subtitle than a main title.

The title has been changed to "New insights into atmospheric iodine gas-to-particle conversion"

Reviewer Comments, second round:

Reviewer #1 (Remarks to the Author):

I looked at the replies, changes and the edited manuscript. In my opinion the authors answered all questions accordingly and made helpful changes to the manuscript. Therefore I recommend publication without further changes and acceptance as it is.

Reviewer #2 (Remarks to the Author):

Gomez Martin et al have revised the manuscript in ways that present a clearer description of the study findings. In particular, the revised title and Figure 7 caption are significantly improved. The remaining minor clarification comments below refer to the line numbers in the track changes manuscript.

Comments:

- The authors note that the mechanism shown in Figure 7 cannot be fully explained in the main text due to length constraints and so Section S6 is primarily used for this purpose, in addition to some added sentences, which are helpful. With this in mind, it would be useful for the authors to refer to Section S6 in the added sentence on Lines 260-262 of the discussion, as well as in the Figure 7 caption. I could not find reference to Section 6 of the supporting information in the main text, but it is possible that I missed it. Regardless, given its importance to the results of the work, it should be referred to where suggested to improve clarity.
- Given the long length of the Supplementary Material, please refer to specific text sections when possible. See Lines 170, 209, 252, 311, 358, and 385, in particular.
- Line 56: Please fix wording, replacing "chemi-ionization sources" with the proper phrasing of "reagent ions" and consider the clearer phrasing "reagent ions for reaction with the analytes".
- Line 607: Fix typo here, as HOIO₂⁺ does not correspond to m/z 144.
- Lines 296-297: The phrase "In the polar MBL, precursor emissions are less intense..." is still missing a reference, even though it is stated that this was addressed.
- Line 300: Fix typo – should be "Arctic".
- Figure 3 caption: It would be useful to state here that HOIO₂ would not be expected to be observed due to the PI of 10.5 eV that was used.

Paper Ref: NCOMMS-20-10505-A

Title: “New insights into atmospheric iodine gas-to-particle conversion ”

RESPONSE TO THE REVIEWERS’ REPORTS

We are grateful to reviewer #2 for helpful and constructive comments and suggestions. We address them point by point below. The Reviewer’s comments are shown in **bold typescript**, our response in normal typescript. Additions to the manuscript are highlighted in red.

Reviewer #2 (Remarks to the Author):

Gomez Martin et al have revised the manuscript in ways that present a clearer description of the study findings. In particular, the revised title and Figure 7 caption are significantly improved. The remaining minor clarification comments below refer to the line numbers in the track changes manuscript.

Comments:

- The authors note that the mechanism shown in Figure 7 cannot be fully explained in the main text due to length constraints and so Section S6 is primarily used for this purpose, in addition to some added sentences, which are helpful. With this in mind, it would be useful for the authors to refer to Section S6 in the added sentence on Lines 260-262 of the discussion, as well as in the Figure 7 caption. I could not find reference to Section 6 of the supporting information in the main text, but it is possible that I missed it. Regardless, given its importance to the results of the work, it should be referred to where suggested to improve clarity.

Sections 1-5 of the Supplementary Information have been transferred to the Methods section of the main text by editorial request (no word limit in Methods). Most of Section 6 has also been transferred to the main text by editorial recommendation. Since the 5000-word limit does not include Methods, there was in fact plenty of room for the discussion of the mechanism in the main text. The remaining text, which mostly concerns interpretation of previous results, which is not essential for the results of this work (last part of section 6 and section 7), has been renamed as Supplementary Notes 1 and 2 by editorial request. Tables S3, S5 and S6 which are directly related to the mechanism and the determination of upper limits for relevant rate constants have also been transferred to the main manuscript to complete a total of 10 allowed display items.

- Given the long length of the Supplementary Material, please refer to specific text sections when possible. See Lines 170, 209, 252, 311, 358, and 385, in particular.

See above.

- Line 56: Please fix wording, replacing “chemi-ionization sources” with the proper phrasing of

“reagent ions” and consider the clearer phrasing “reagent ions for reaction with the analytes”.

Done

- Line 607: Fix typo here, as HOIO₂⁺ does not correspond to m/z 144.

Done.

- Lines 296-297: The phrase “In the polar MBL, precursor emissions are less intense...” is still missing a reference, even though it is stated that this was addressed.

We apologize for forgetting to include the requested reference, which has been inserted in the revised version of the manuscript. The sentence was in fact slightly misleading and we have rewritten it as follows: **In the polar MBL, active iodine (IO_x) mixing ratios are in general lower than in macroalgae-rich mid-latitude locations⁵, and consequently lower IxOy mixing ratios should result in slower IOP formation rates (Table S8).**

-Line 300: Fix typo – should be “Arctic”.

Done.

- Figure 3 caption: It would be useful to state here that HOIO₂ would not be expected to be observed due to the PI of 10.5 eV that was used.

Added in the figure caption of Figure 3: **In this experiment, HOIO₂ would not be expected to be observed due to the PI energy of 10.5 eV below the ionization threshold.**